# High-Entropy Materials in SOFC Technology: Theoretical Foundations for Their Creation, Features of Synthesis, and Recent Achievements

**DOI:** 10.3390/ma15248783

**Published:** 2022-12-08

**Authors:** Elena Y. Pikalova, Elena G. Kalinina, Nadezhda S. Pikalova, Elena A. Filonova

**Affiliations:** 1Laboratory of Solid Oxide Fuel Cells, Institute of High Temperature Electrochemistry, Ural Branch of the Russian Academy of Sciences, Yekaterinburg 620137, Russia; 2Department of Environmental Economics, Graduate School of Economics and Management, Ural Federal University, Yekaterinburg 620002, Russia; 3Laboratory of Complex Electrophysic Investigations, Institute of Electrophysics, Ural Branch of the Russian Academy of Sciences, Yekaterinburg 620016, Russia; 4Department of Physical and Inorganic Chemistry, Institute of Natural Sciences and Mathematics, Ural Federal University, Yekaterinburg 620002, Russia; 5Institute of Metallurgy, Ural Branch of the Russian Academy of Sciences, Yekaterinburg 620016, Russia

**Keywords:** solid oxide fuel cells (SOFC), high-entropy alloys, high-entropy oxides, cathodes, anodes, solid electrolytes, electrical conductivity, electrochemical activity, maximum power density

## Abstract

In this review, recent achievements in the application of high-entropy alloys (HEAs) and high-entropy oxides (HEOs) in the technology of solid oxide fuel cells (SOFC) are discussed for the first time. The mechanisms of the stabilization of a high-entropy state in such materials, as well as the effect of structural and charge factors on the stability of the resulting homogeneous solid solution are performed. An introduction to the synthesis methods for HEAs and HEOs is given. The review highlights such advantages of high-entropy materials as high strength and the sluggish diffusion of components, which are promising for the use at the elevated temperatures, which are characteristic of SOFCs. Application of the medium- and high-entropy materials in the hydrocarbon-fueled SOFCs as protective layers for interconnectors and as anode components, caused by their high stability, are covered. High-entropy solid electrolytes are discussed in comparison with traditional electrolyte materials in terms of conductivity. High-entropy oxides are considered as prospective cathodes for SOFCs due to their superior electrochemical activity and long-term stability compared with the conventional perovskites. The present review also determines the prioritizing directions in the future development of high-entropy materials as electrolytes and electrodes for SOFCs operating in the intermediate and low temperature ranges.

## 1. Introduction

Solid oxide fuel cells (SOFCs) are highly efficient electrochemical devices that directly convert fuel energy into electricity in a manner that is ecologically friendly. As such, they have attracted much attention from scientific groups and manufacturers [1]. In order to optimize the efficiency of SOFCs and extend their lifetime, increase the types of fuel that can be used, reduce operating temperatures [2], expand the possibility of their use in multigeneration [3,4] and hybrid [5] systems, and successfully market them [6], new advanced SOFC materials must be developed, as well as cost-effective methods of fabrication [7,8]. Currently, a new concept for designing materials based on the possibility of regulating the phase stability and functional properties of multicomponent solid solutions through the control of configurational entropy finds its application in various areas of science and technology [9,10,11]. New high- and medium-entropy materials have recently been developed for SOFC electrolytes [12,13,14], cathodes [15,16,17], anodes [18], and protective coatings for interconnectors [19]. The aim of this review is to briefly describe the theoretical foundations for the creation of high-entropy materials, discuss features of their synthesis, and provide an overview of recent achievements and their possible application in SOFC technology and related fields.

## 2. High-Entropy Alloys: Theory, Achievements, and Prospects for Use in SOFC Technology

### 2.1. Theoretical Founditions and Recent Advances in the Development of High-Entropy Alloys

Recently, in the field of materials science a new direction has been widely developed which is related to the study of the unusual properties of high-entropy alloys (HEAs), first proposed in 1995 by Yeh and Huang [20]. Usually, such materials consist of several elements (at least five) taken in an equimolar ratio, for example, the CrMnFeCoNi system, obtained by Cantor et al. [21], and high-entropy CuCoNiCrAlxFe alloys, produced by the Yeh et al. group [22]. Unlike conventional multicomponent alloys, which are mainly multiphase structures, these HEAs were characterized by the appearance of a single-phase solid solution with a uniform and disordered distribution of their constituent elements. The obtained HEAs [21,22] demonstrated structural stability, including at elevated temperatures, high strength while maintaining plasticity, low thermal conductivity, slow diffusion of elements in the alloy, and grain boundary mobility. It was found that the structure of the HEA acquires a noticeable distortion of the crystal lattice which hinders the movement of dislocations and leads to enhanced hardness. Slow diffusion (sluggish diffusion) of elements in HEAs is associated with the occurrence of local stresses and an increase in the migration energy barrier. Slowing down the diffusion processes in HEAs can contribute to the formation of a nanosized grain structure [21]. Due to the distortion of the crystal lattice and mixing of the components, a “cocktail” effect occurs, in that the properties of the resulting material differ significantly from the properties of the individual components included in its composition [23].

To date, hundreds of HEA compositions with a variety of structures have been studied, namely, single-phase solid solutions with face-centered cubic (FCC), body-centered cubic (BCC), hexagonal close-packed (HCP), and orthorhombic structures, as well as HE amorphous metallic glasses [24] (Figure 1a). The main features of HEA’s properties (summarized in Figure 1b) and the scope of their application are presented in a number of recent reviews [10,11,24,25,26,27,28,29,30].

A number of alternative names for HEAs have been proposed in the literature, such as multi-principal element alloys (MPEAs) [31] and complex concentrated alloys (CCAs) [32]. In addition, the concept of medium-entropy alloys (MEAs) was introduced for alloys containing four or fewer elements in an equiatomic ratio or close to it [33]. In analyzing the place of HEA among other types of condensed matter, Gelchinski et al. [10] suggested that HEAs can be placed between amorphous compounds and nanocrystals. From one perspective, no regions with a repeating unit cell can be found in their structure in contrast to nanocrystals, where such regions, albeit nanosized, are present. Alternatively, the degree of order in HEAs is higher than in amorphous materials since there are regularly located lattice sites (according to the type of crystal lattice) with disordered atoms of various elements.

Among the most used HEA components are Fe, Ni, Co, Cr, Al, Cu, Mn, and Ti, which are also included in the classical compositions (Figure 2).

The basic principle of the synthesis of high-entropy materials is the formation of a homogeneous solid solution from a mixture of components. For example, the transition from a multi-phase to a single-phase system is possible with rapid cooling of the mixture [34]. In this case, a disordered substitutional solid solution is formed, in which atoms are randomly located at the lattice sites, which leads to an increased configurational entropy of the resulting phase. The difference between the atomic radii of the initial HEA components distorts the lattice and the structure of such phases can be considered, as was mentioned above, as an intermediate state between stable crystalline phases and metastable amorphous materials in which there is no long-range order. This raises the question of the long-term stability of the resulting homogeneous solid solution in a high-entropy state and the possible occurrence of local ordering with time.

Thermodynamic stability conditions lead to the requirement for a negative change in the Gibbs energy during the formation of a given material (at constant pressure and temperature):(1)ΔG=ΔH−TΔS<0,
where ΔG is a Gibbs-free energy change, ΔH is an enthalpy change, ΔS is an entropy change, and T is an absolute temperature. The entropy of an alloy comprises both the configurational entropy and the thermal entropy. The configurational entropy, also called as entropy of mixing, $\Delta S_{mix}$, in a multicomponent equiatomic system depends on the number of components as follows:(2)$$\Delta S_{mix}=R\;\ln\left(N\right),$$
where R is the absolute gas constant, and N is the number of the components.

Figure 3a shows the entropy of mixing, calculated according to Equation (2) depending on the number of elements in equimolar alloys [30]. Thus, for equimolar two-, four-, and five-component alloys, the entropy of mixing was equal to 5.76, 11.52, and 13.37 J/K mol, respectively. Based on the entropy approach, alloys can be roughly divided into three categories according to their entropy of mixing on the supposition of random distribution of elements in a solid solution, namely (i) low-entropy alloys (traditional alloys) with one or two main elements with $\Delta S_{mix}\leq 5.76\;$ or 0.69 R, (ii) medium-entropy alloys with two to four base elements with entropy of mixing in the range of $0.69\;R<\Delta S_{mix}<1.61\;R$, and (iii) high-entropy alloys with five or more base elements with $1.61\;R\leq \Delta S_{mix}$, as shown in the inset in Figure 3a.

An increase in the number of components in an alloy increases the configurational entropy, which reduces the Gibbs energy, which is also facilitated by an increase in temperature. The competing factor is the change in enthalpy; however, at a high entropy of mixing and a total negative change in the Gibbs energy, a homogeneous disordered solid solution is formed instead of a multiphase system. In this case, possible processes of spinodal decomposition, the formation of intermetallic compounds, and the appearance of ordered structures and secondary phases are suppressed. It is also noted in [30] that in HEAs the percentage of individual components in the HEA can vary from 5 to 35%, which significantly expands the range of materials under consideration. It is worth noting the surprising result obtained by Senkov et al. in [31]. The authors evaluated 134,547 alloy systems using a calculated phase diagram (CALPHAD) method and found that the formation of solid solution (SS) or intermetallic alloys (IM) became less likely as the number of elements increased, while the probability of the formation of mixed SS + IM phases increased (Figure 3b). As the number of elements increases, $\Delta S_{mix}$ rises slowly while the probability of at least one pair of elements favoring IM formation increases more rapidly, explaining this apparent contradiction with the major principle of HEAs.

The following quantitative empirical factors (Ω and δ criteria) determine the stability of HEAs:(3)$$Ω=\frac{T\Delta S_{mix}}{\left|\Delta H_{mix}\right|}$$
(4)$$\delta =\sqrt{\sum_{1}^{n}c_{i}{\left(1-r_{i}/\bar{r}\right)}^{2}},$$
where $\bar{r}=\sum_{1}^{n}c_{i}r_{i}$, $\Delta S_{mix}=-R\sum c_{i}lnc_{i}$ is a mixing entropy; $\Delta H_{mix}$ is a mixing enthalpy; $c_{i}$ and $r_{i}$ are a concentration and an atomic radius of the *i*-th element, respectively; and $\bar{r}$ is an average atomic radius. The Ω value characterizes the ratio between the mixing entropy and the enthalpy change, while the δ value characterizes the difference in the atomic radii of the elements included in the HEA. The Ω and δ criteria values equal to 1.1 and 6.6 were determined in [35] that ensured stability of the TiZrNbMoVx and CoCrFeNiAlNbx alloys.

The difference in the values of the electronegativity of the elements included in the HES was also considered as a stability factor (Δχ criterion):(5)$$\Delta \chi =\sqrt{\sum_{1}^{n}c_{i}{\left(1-\chi_{i}/\bar{\chi }\right)}^{2}},$$
where $\bar{\chi }=\sum_{1}^{n}c_{i}\chi_{i}$, $\chi_{i}$ is the electronegativity of the *i*-th element, and $\bar{\chi }$ is an average electronegativity. The low value of the Δχ criterion ensures a uniform ability to attract electrons across the lattice, leading to a stable solid solution [36]. Polletti and Battezzati assigned the values of Δχ<6% and δ<6% as a guideline for the selection of elements to predict new HEAs [37]. The stability of a high-entropy state in HEAs is also affected by such parameters as the density of valence (VEC) and free electrons (e/a), i.e., their number per atom [37,38].

In addition to the configuration component, the total value of entropy also includes temperature-dependent contributions, vibrational, electron, and magnetic. Using finite-temperature ab initio methods, Ma et al. [39] revealed that entropy contributions beyond the configurational contribution were crucial for determining phase stability (for hcp, fcc, and bcc structures) and such properties of the Cantor alloy (CoCrFeMnNi) as thermal expansion and bulk modulus. According to their results, electronic and magnetic entropies can contribute up to 50% of the configurational entropy value. Shuo Wang et al. [40], when considering electronic and thermodynamic properties of this alloy in nonmagnetic (NM) and ferrimagnetic (FIM) states, demonstrated that the magnetic properties of the constituent atoms play an important role in the thermodynamics of this HEA. Calorimetric measurements of thermal entropy of the series of the HEAs, including the Cantor alloy by Haas et al. [41], showed that it was a few folds higher than the configurational contribution. However, for alloys with the same crystal structure, the thermal contributions did not depend on the number and concentration of the alloying elements; therefore, it was directly demonstrated that despite the thermal entropy being significantly higher than the configurational entropy, it does not increase the thermal stability of HEAs relative to simple alloys or pure metals.

It is worth mentioning that an increasing number of recent studies have been revealing that the formation of single-phase solid solutions in HEAs show a weak dependence on maximization of $\Delta S_{mix}$ through equiatomic ratios of elements. Moreover, the entropy maximum has been shown to be not the most significant parameter in the creation of multicomponent alloys with high functional properties. On the basis of these studies, the recent review by Li and Raabe [42] demonstrated that by changing the strategy of creating alloys from single-phase equiatomic to two- or multi-phase non-equiatomic, it is possible to obtain high-strength and superplastic HEAs.

### 2.2. HEA Synthesis Methods

A wide range of synthesis techniques has been developed for HEAs. Methods of synthesis, both bulk and powder-based, have been reviewed in a number of review papers [10,43,44]. The most commonly used solid-state method to obtain HEA powders is that of metal alloying as it is the most simple and most suitable method due to its increased solid solubility, high level of homogeneity, and room-temperature processing [45]. The method includes high-energy planetary ball milling (usually using stainless steel balls) in a protective milling media (Figure 4a). Mechanical alloying is followed by the consolidation of HEAs using, for example, spark plasma sintering (SPS) [46]. The second most commonly used method to obtain full pre-alloyed HEA powders suitable, for example, for additive manufacturing is that of atomizing, using different gas or liquid streams (Figure 4b) [47]. HEAs can also be manufactured via liquid-state synthesis methods using melting and casting techniques such as conventional arc melting [48,49] (Figure 4c), as well as vacuum induction melting, directional solidification, infiltration (Figure 4d), etc. Gas-state HEAs synthesis methods, through the deposition of thin HEAs films, include the plasma spray process (Figure 4e), thermal spraying, and magnetron sputtering (Figure 4f), molecular beam epitaxy, vapor deposition, etc. More detailed descriptions of liquid and gas-state methods are given in the recent reviews [50,51]. Among the most cost-effective methods to form HEA coatings is electrochemical deposition which, it should also be noted, does not require complicated equipment [52]. This provides the opportunity to control the film thickness and content simply by regulating the deposition parameters such as current density and applied potential.

Nanosized HEAs are reported to have superior performance in the fields closely related to the fuel cell technology, namely catalysis, energy storage, and conversion [28,53,54]. However, their production is challenging due to the critical conditions that are required for the synthesis, such as heating up to extremally high temperatures followed by rapid cooling to “freeze” the nonequilibrium state. There are a few synthesis techniques which allow HEA nanoparticles to be obtained: a carbothermal shock technique [55] (Figure 4g), sputter deposition [56], solvothermal synthesis [57], and fast-moving bed pyrolysis method [58] (Figure 4h), a microwave heating method that utilizes carbon substrates [59], etc.

Kinetically controlled laser synthesis was reported as a new, highly reproducible, and scalable method to obtain HEA’s nanoparticles. Due to fast kinetics, it allows the formation of a large number of isolated ultrafine nanoparticles with properties close to those of the ablation target used [60]. Recently, Wang et al. proposed using laser scanning ablation as a simple and general approach to synthesizing both high-entropy alloy and ceramic nanoparticles [61]. The advantage of this method is that it can be implemented at atmospheric temperature and pressure, both for synthesis where there is a substrate or where no substrate is used. The ultrarapid process ensures the synthesis of HEA from up to nine metallic elements regardless of their thermodynamic solubility.

### 2.3. HEAs Applications

Although conventional alloys are presently used in modern advanced applications, it should be noted that the development of new alloys based on one or two major elements gradually approached the limit of feasible combinations at the end of the twentieth century [62]. This saturation created certain difficulties in meeting material requirements in the face of the anticipated technology-driven performance leap. Under these circumstances, HEA and related materials can provide new advanced features. Studies undertaken in HEA high temperature applications have shown that appropriate composition design and process selection can lead to HEA replacing traditional alloys for such energy-related applications as energy conversion and storage [63], hydrogen storage [27,64], catalysis [28], electrocatalysis [65,66], electrocatalysis for hydrogen evolution, oxygen evolution, and oxygen reduction reaction [67,68], surface electrocatalysis [69], nuclear power [70,71,72], lithium and sodium batteries [73,74], and coatings for energy applications [75].

#### 2.3.1. SOFC Interconnectors

There are almost no studies concerning HEAs application in the fuel cell technology including SOFCs. However, due to their high heat resistance and superior oxidation and corrosion resistance, these materials can reasonably be considered for future use in creating protective layers for SOFC interconnectors. Interconnectors in SOFCs serve to multiply the cell voltage output by adjusting unit cell elements in electrical series and separating fuel from oxidant [76], which is schematically illustrated in Figure 5. In order to perform their intended functions, interconnects should have the following characteristics [77]:(1)High electrical (preferably electronic) conductivity under the SOFC operating temperatures (minimum value of 1 S/cm);(2)Sufficient chemical and phase, as well as microstructural stability in both reducing and oxidizing atmospheres;(3)Gas-tightness to prevent direct combination of oxidant and fuel during fuel operation;(4)Good thermal conductivity (minimum value of 5 W/(m K);(5)Thermomechanical compatibility with electrodes and electrolytes (Thermal expansion coefficient (TEC) value close to (10 × 10^−6^ 1/K in the 25–1000 °C range) to minimize thermal stresses during stack startup and shutdown;(6)No reaction or inter diffusion between interconnectors and its adjoining components (anode and cathode);(7)Excellent resistance to oxidation, sulfidation and carbonization, and adequate strength and creep resistance at elevated temperatures;(8)Easy fabrication and low cost.

The interconnects used in the modern SOFC technology can be divided into two categories, ceramic and metallic interconnectors. Accepter-doped lanthanum chromites with a perovskite structure are usually used as ceramic interconnectors. However, at temperatures below 800 °C, their electrical conductivity experiences a substantial decline. Among the major obstacles to mass production of doped LaCrO_3_–ceramic interconnectors is their inferior sintering behavior in air atmosphere due to easy Cr (VI) volatilization, making it problematic to achieve the necessary gas-tightness. Moreover, the high oxygen diffusivity in a reducing atmosphere enhances the oxygen transport through the interconnectors which may deteriorate the fuel cell performance. Compared with ceramic interconnectors, metallic interconnectors exhibit better manufacturability and lower cost, possess improved mechanical strength, and demonstrate higher electrical and thermal conductivity [78]. Cr-based oxide-dispersion-strengthened alloy, particularly Ducrolloy (Cr–5Fe–1Y_2_O_3_) is considered to be a suitable replacement for ceramic interconnectors is SOFCs with decreased operating temperatures. Fe–Cr-based alloys, specifically ferritic stainless steels with an optimum Cr-content of 17–25% demonstrate better workability and lower cost compared with Cr-based alloys. Moreover, the body-centered cubic structure of ferritic stainless steel provides thermomechanical compatibility with other SOFC components. Ni–Cr-based alloys exhibit excellent oxidation resistance and high electrical conductivity; however, they have an unacceptable high TEC value. Therefore, ferritic stainless steel is the most promising candidate to be used as interconnectors in intermediate and low temperature (IT and LT) SOFCs.

Despite all the advantages of Cr-containing alloys, they present two major limitations. Under SOFC operation, oxidation of the ferritic stainless steel interconnector leads to the formation of a Cr_2_O_3_ scale that increases the electrical resistivity and thus lowers the stack efficiency. Chromium, evaporated from the interconnector surface as chromium oxyhydroxide CrO_2_(OH)_2_ and chromium oxide (CrO_3_) can deposit onto the cathode, deteriorating its electrochemical activity.

To overcome these drawbacks, metallic interconnectors are coated with conductive ceramics to improve their surface stability. Perovskites and spinels have been considered as the prime candidates for diffusion barrier layers for interconnectors based on the need for high conductivity [79]. The main restriction of perovskites coatings is the difficulty of obtaining good compactness and adhesion, while the preparation of a spinel coating is more realizable and can be achieved by either depositing a spinel oxide or thermally converting from an alloy coating. Numerous investigations have indicated that the desired electrical conductivity and TEC of spinel can be obtained through proper compositional modification [80]. Mn-Co and Mn-Cu systems possess superior electrical properties, while ferrite spinel have a TEC value (10.8 × 10^−6^ 1/K) close to ferritic stainless steel. Spinels that are highly doped by various cations can contribute to better performance. Zhao et al. [81] showed that FeCoNi (1:1:1) coating deposited on SUS 430 steel by magnetron sputtering and then thermally converted to protective and electrically conductive spinel coating, significantly suppressed the growth of Cr_2_O_3_, reduced the oxidation rate of substrate steel, and reduced the area specific resistance of the surface scale. In their recent study, the same authors extended the alloy composition to Fe:Co:Ni:Mn:Cu = 1:1:1:1:1:1 [19]. The sputtering target with a designed composition was obtained via vacuum induction melting. The coating of 4 μm in thickness was deposited on the SUS 430 steel substrate under sputtering power of 1.2 kW for 2 h at 200 °C in a chamber with an Ar pressure of 0.13 Pa. The coating exhibited a single fcc structure of γ-phase. To oxidize the coating, the samples were heated to 800 °C and held at this temperature for one week, then cooled down to room temperature. This cycle was repeated 10 times; therefore, the total oxidation time was 10 weeks (1680 h). Figure 6 demonstrates cross-sectional images of the coating after 1 (Figure 6a), 5 (Figure 6b), and 10 (Figure 6c) weeks of thermal cycling. After being exposed to air, the alloy coating was converted into a high-entropy spinel coating of (FeCoNiMnCu)_3_O_4_ with a thin layer of CuO on the top after the first week (Figure 6a) and subsequently a uniform (FeCoNiMnCu)_3_O_4_ coating was formed by means of a solid-state reaction after 10 weeks. The interface substrate/coating Cr-rich oxide layer of 9 μm in thickness was formed after the first week, after 5 weeks it increased up to 11 μm, and up to 12 μm after 10 weeks, while Cr concentration on the surface was only 4.34% after 10 weeks oxidation (Figure 6c). As a result, the high-entropy spinel coating thermally converted from the HEA coating showed it was capable of effectively blocking Cr out-diffusion. The values of the area-specific resistance (ASR) measured at 800 °C was 8.21, 7.67, and 6.59 mΩ·cm^2^ at 800 °C after 1, 5, and 10 weeks of oxidation, respectively. A decrease in ASR can be related to the conversion of (FeCoNiMnCu)_3_O_4_/CuO structure into a uniform (FeCoNiMnCu)_3_O_4_ spinel layer. The scale ASR was 17 mΩ·cm^2^ at 600 ℃, which is well below the accepted ASR limit (0.1 Ω·cm^2^). This suggests the FeCoNiMnCu coating is very promising for the SOFC interconnectors with lower exposure temperature. It should be also noted that the ASR values obtained for the HEA coating are significantly lower than those obtained for multicomponent spinel coatings MnCu_0.25_Fe_0.25_Ni_0.5_CoO_4_ and MnCu_0.5_Fe_0.25_Ni_0.25_CoO_4_ (0.134 and 0.180 Ω cm^2^ at 750 ℃) [82]. The conductivity of (FeCoNiMnCu)_3_O_4_ spinel was not measured; however, the electrical properties of a number of HE spinels were evaluated in a number of studies. Three different oxides, namely, (CoCrFeMnNi)_3_O_4_, (CoCrFeMgMn)_3_O_4_, and (CrFeMgMnNi)_3_O_4_ were obtained as single-phase materials using the solid-state reaction route in [83]. Starting oxides were mixed in a planetary mill at a rotating speed of 600 rpm for 90 min, pressed into pellets, calcined at 1000 °C in air for 20 h and then rapidly cooled to room temperature. All the obtained materials possessed cubic structure (sp. gr. Fd-3m). The TEC values for the above-mentioned compounds, evaluated from the high-temperature XRD study, were 9.7, 8.8, and 9.8∙10^−6^ 1/K, respectively. The Seebeck coefficient value was within −90 to −150 μV/K, which is typical for the semiconductors, in which electrons are the main charge carriers. The conductivity value was approximately 145 S/cm for Co-containing and 7 S/cm for Co-free spinels at 600 °C.

It is worth mentioning that lithiated HE spinels are successfully used as electrodes in lithium batteries (LIBs), as the spinel structure is characterized by its unique three-dimensional Li^+^ transport pathways. Transition metal cations are randomly dispersed in the A- and B-sites of a HE spinel lattice, thus resulting in various valence states of the cations and plenty of oxygen vacancies. The oxygen vacancies can increase the configurational entropy and promote Li^+^ transport as well, ensuring superionic Li^+^ mobility (>10^−3^ S/cm) at room temperature [84]. (Mg_0.2_Ti_0.2_Zn_0.2_Cu_0.2_Fe_0.2_)_3_O_4_ was reported in [85] as an anode for LIBs with fast reaction kinetics, delivering a reversible capacity of 504 mA h/g at a current density of 100 mA/g after 300 cycles and a rate capability of 272 mA h/g at 2000 mA/g. Non-equimolar HE spinel ($\Delta S_{mix}=1.5\;R$) with elemental concentrations of Co, Cr, Fe, Mn, and Ni equal to 0.2, 11.2, 8.0, 28.6, and 22.0%, respectively, was reported in [86] as a novel anode material for LIBs, demonstrating superior charge–discharge capacity of 1235 mA h/g with 90% capacity retention after 200 cycles and rate capability of 500 mA h/g at 2000 mA/g.

#### 2.3.2. HEAs Application in Hydrocarbon-Fueled SOFCs

Due to relatively high operation temperatures and an all-solid design, SOFCs are highly suitable for the use of hydrocarbons (HCs) for auxiliary power units (APU) for vehicles as well as for various stationary applications [87]. While employing methane as the fuel, it is generally assumed that SOFCs are operating on syngas (CO+H2) formed during external, indirect internal, or internal steam reforming of methane (SRM) (Figure 7). The state-of-the-art catalysts for SRM utilize precious metals such as Pt and Rh, which are considerably expensive. Varying compositions can modulate the absorption energy of the alloys, which provides an innovative way to reduce the cost of catalytic materials by adding low-cost materials such as diluents [88,89]. The electrochemical catalytic activity of a number of such compositions was studied recently: CoCuGaNiZn [88], AgAuCuPdPt [90], CrMnFeCoN [91], and AlNiCuPtPdAu [92].

The direct use of HCs can enhance fuel cell efficiency through the minimizing the losses of efficiency involved by using the external reformers; however, HCs cannot be utilized directly in SOFCs with commonly used Ni-based cermet due to their carburization, which can be described as follows [87]: (1) carbon from HC deposits on the Ni surface; (2) it undergoes dissolution in the bulk of the Ni particles; and (3) it is precipitated as a fiber. This leads to Ni loss as Ni atoms separate from the surface with the growing carbon fibers. The microstructure damage leads to a reduction in the conductivity. Additionally, as a result of the growth of carbon fibers, mechanical stresses are generated in the SOFC, resulting in its fracture [88]. The deposition of a catalyst layer with high catalytic activity for HCs over the nickel-based anode was demonstrated to be an effective way of enhancing the performance and long-term stability of HC-fueled SOFCs [89]. An alternative approach is the development of hydrocarbon compatible SOFC anodes [87]. Among those which are suitable for the HC-fueled SOFCs, are (a) various ceramic anodes with perovskite (chromites, titanates, vanadates, manganates, molybdates, cerates/zirconates, etc.), double perovskite, pyrochlore, spinel, rutile structures; (b) ceria and doped ceria with a fluorite structure, (c) Ni, Co, Cu-based cermets with the addition of oxygen ionic (YSZ, ScSZ, doped CeO_2_) or protonic (cerate-zirconate) conductors d) Ni-Cu, Ni-Fe, Co-Cu, Fe-Cu, Ni-Mo bimetallic cermets. The near equimolar and non-equimolar HEAs having five or more transition metals in its content along with their mingled sites over the surface are perspective materials for SOFC cermet anodes [93,94].

The HEA-based anode and reforming catalyst containing Cu, Ni, Co, Fe, and Mn were synthesized on supported Gd-doped CeO_2_ (GDC) and evaluated for SRM under SOFC operating conditions in [95]. HEA was prepared using the co-precipitation method from related nitrate precursors using citric acid as a chelating agent. Calcination was performed at 500 °C for 6 h with a heating/cooling rate of 5 °C/min. Then, the obtained HEA was mixed with a commercial GDC powder in a wt. ratio of 65:35. For the sake of comparison, Ni/YSZ (65:35) and Ni/GDC (50:50) were prepared. The metal content in HEA-GDC (wt.%) was: 9.75 (Ni), 13 (Co), 16.25 (Cu), 16.25 (Fe), and 9.75 (Mn). The catalytic activity of SRM catalysts was examined using a fixed-bed tube reactor at 700, 750, and 800 °C at a gas hourly space velocity (GHSV) of 45,000 1/h. The lowest reforming rate was observed for HEA/GDC, which increased from 27% at 700 °C to 35% at 750 °C and then to 42% at 800 °C, accompanied by an increase in hydrogen yield with temperature (Figure 8a). Both Ni/YSZ and Ni/GDC catalysts showed a high reforming rate of ∼3.5 mol_CH4_/mol_Ni_ 1/s at 750 °C, while the HEA/GDC catalyst showed a reforming rate of ∼1 mol mol_CH4_/mol_HEA_ 1/s. However, both Ni-based catalysts rapidly degraded with time, while the HEA/GDC catalyst demonstrated excellent stability (Figure 8b). Electrochemical activity, stability, and carbon-resistance of the HEA/GDC anode catalyst were investigated in a SOFC cell test using a HEA/GDC-Ni/ScSZ|ScSZ|LSM/YSZ cell (Figure 8c).

A current density of ∼100 mA/cm^2^ was achieved at 750 °C. The non-ohmic resistances of the cell decreased after 15 h indicating higher mass transfer due to an increase in the reforming rate, while the ohmic resistance remained stable. After the 30 h cell test, the anode layers were analyzed using SEM microscopy and Raman spectroscopy. The presence of carbon on Ni/YSZ anode was 17.3%, while the surface of the post-test HEA/GDC anode layer was carbon-free. The absence on the Raman spectrum of D and G bands at 1345 and 1595 1/cm, respectively, suggested that the HEA/GDC anode was free of both amorphous and graphitic-typed carbons. Promising results obtained in [95] offer prospects for further study and development of carbon resistant anodes for HC-fueled SOFC with direct internal reforming.

## 3. High-Entropy Ceramic Materials: Theoretical Aspects, Synthesis, Applications, and Use in SOFC Technology

The approaches developed for HEAs were later applied to create high-entropy ceramic materials. Inspired by research activities in the metal alloy communities and fundamental principles formulated by Navrotsky and Kleppa [96] on entropy regulation of the normal-to-inverse transformation in simple spinels, Rost et al. [97] validated “the entropy ansatz” on a mixture of MgO (Rock-salt (RS) structure), CoO (RS), NiO (RS), CuO (Tenorite structure), and ZnO (Wurtzite structure). Oxides were ball-milled for 2 h, compacted, and calcined at 700–1100 °C with following quenching to room temperature. It was found that full conversion to a single-phase rock-salt quinary high-entropy oxide (HEO) (Mg_0.2_Co_0.2_Cu_0.2_Ni_0.2_Zn_0.2_)O formed above 850 °C. The random distribution of cations on the cation sublattice sites was confirmed by extended x-ray absorption fine structure (EXAFS). The authors proved the reversibility of the entropy-driven transition in the 1000–750–1000 °C cycling. Considering the influence of composition variations, they showed that removing any of the components resulted in a multi-phase solid solution. Varying the concentration of single components in the equimolar composition by ±2, ±6, and ±10%, they observed that the transition temperature was increasing due to a decrease in the configurational entropy. Later, a reversible multi-to-single-phase transition was observed for (Gd_0.2_La_0.2_^Nd^_0.2_Sm_0.2_Y_0.2_)MnO_3_ and (Gd_0.2_La_0.2_^Nd^_0.2_Sm_0.2_Y_0.2_)(Co_0.2_Cr_0.2_Fe_0.2_Mn_0.2_Ni_0.2_)O_3_ HEOs of perovskite [98] and (Ce_0.2_Zr_0.2_Hf_0.2_Sn_0.2_Ti_0.2_)O_2_ of a fluorite [99] structure, indicating entropy stabilization of the compounds. These experiments created a solid base for further development of new family of high-entropy oxides (HEOs) which are currently attracting significant interest due to their unique structural characteristics and correlated possibilities for tailoring the functional properties [9,100,101,102]. Figure 9 summarizes the number of publications concerning different types of HEOs that have appeared since 2010 and advanced applications of HE ceramic materials in energy-related fields.

### 3.1. Entropy Stabilization Approach to the Creation of HEOs

The general concept of entropy stabilization of the oxide system is based on the possibility of stabilizing a single-phase crystal structure by increasing the configurational entropy ($S_{conf}$) of the system, which can be achieved by increasing the number of elements. Molar $S_{conf}$ can be calculated as follows:(6)$$S_{conf}=-R\;\left[{\left(\overset{N}{\underset{i=1}{\sum }}x_{i}lnx_{i}\right)}_{cation-site}+{\left(\overset{M}{\underset{j=1}{\sum }}x_{j}lnx_{j}\right)}_{anion-site}\right]$$
where $x_{i}$ and $x_{j}$ represent the mole fraction of elements in the cation and anion sites, respectively. In addition, in the case of two cation sublattices in the ceramics, such as perovskite, spinel, pyrochlore oxides, etc., Equation (6) should be modified as follows:(7)$$S_{conf}=-R\left[{\left(\sum_{h-1}^{M}x_{h}lnx_{h}\right)}_{A-site}+{\left(\sum_{i-1}^{N}x_{i}lnx_{i}\right)}_{B-site}+{\left(\sum_{j-1}^{L}x_{j}lnx_{j}\right)}_{anion-site}\right]\;$$

For HEOs, the contribution of the anion sublattice, represented by only one type of atom, oxygen, to entropy stabilization of the structure is expected to be minor (although there are also the cases of anion-substituted compounds, for example, with fluorine [104]). However, it should be noted that the anions screen the metal cations and make the introduction of configurational disorder easier, thus increasing the number of possible single phases and stability ranges [97]. Moreover, the anion sublattice can deform to accommodate the differences in cation sizes and bonding, favoring a cation sublattice with as little distortion as possible [105,106]. Based on the classification introduced for HEAs (see Section 2.1), materials with $1.5\;R\leq S_{conf}$ are considered as high entropy, for $1\;R\leq S_{conf}<1.5\;R$ as medium entropy, and $S_{conf}<1\;R$ as low entropy. In many cases, a single-phase state can be achieved at the condition of $1.5\;R\leq S_{conf}$, as then the $T\Delta S_{\mathrm{mix}}$ (Equation (1)) can be high enough to dominate the free energy landscape and overcome $\Delta H_{\mathrm{mix}}$. However, in many cases, an increase in $S_{conf}$ is not enough to compensate for $\Delta H_{\mathrm{mix}}$ and, as a result, intermediate products with more favorable formation enthalpies are produced [9].

In addition to the mixing entropy factor, other mechanisms of stabilization of the single-phase state can also arise in high-entropy oxides, such as the effect of the size and charge state of cations for example, as it was shown in [107] that the presence of Ce^4+^ cations played a crucial role in stabilizing the multicomponent rare earth oxides (REOs) into a phase pure structure. In contrast to HEAs, another critical criterion for HEOs is the role of defect chemistry, which is closely related to the crystal structure. Different from the mixing of multiple elements in HEAs, the “high-entropy mixing” behavior in HECs can be described as the mixing of multiple species which include cations, anions, and defects. This feature was demonstrated in the example of the classic LaMnO_3_ (LMO) perovskite in [108]. LMO was modeled using the compound energy formalism [109] to describe the mixing of ions and vacancies on three sublattices, (La^3+^,Mn^3+^,Va)_1_(Mn^2+^,Mn^3+^,Mn^4+^,Va)_1_(O^2−^,Va)_3_, where Va denotes a vacancy and subscripts 1, 1, and 3 denote the number of site for each sublattice, respectively. The authors noted that, in addition to the composition of high-entropy ceramics, such parameters as oxygen partial pressure and temperature affect the mixing effect, i.e., on the change in entropy, enthalpy, and Gibbs energy. Using the computational method CALPHAD and the chemistry of defects, the values of the thermodynamic parameters of the LMO material were determined by varying the partial oxygen pressure pO_2_ in the range from 10^−20^ to 1 atm, the temperature from 700 °C to 1400 °C, and the A:B ratio from 0.9 to 1.1 in the composition of the perovskite material ABO_3_. It was shown that the mixing of cations in one position affected the mixing of cations in another position, and the electrical neutrality condition imposed an additional condition on the change in thermodynamic parameters during mixing. Mixing effects determine, for example, the type and magnitude of the electrical conductivity of the material, which is important from the point of view of the use of high-entropy ceramics in the creation of SOFCs. The authors conclude that thermodynamic optimization of the mixing effects allow one to obtain the desired character of the mixed oxygen ion and electronic conductivity (MIEC) of the oxide ceramic material.

Recently, Wright et al. proposed that HE ceramic can be expanded to compositionally complex (CCC) or multi-principal cation ceramics (MPCCs) by including medium-entropy and/or non-equimolar compositions [110,111].

### 3.2. Production Methods for the Formation of HEO Materials, Bulk Ceramics and Films

The (Mg_0.2_Co_0.2_Cu_0.2_Ni_0.2_Zn_0.2_)O solid solution, the first known high-entropy (HE) metal oxide, was synthesized by Rost et al. in [97] from the related commercial oxides using a conventional solid-state reaction (SSR) method, which includes the steps of ball-milling and calcination. To facilitate interaction, the materials were pressed into the pellets using a uniaxial hydraulic press at 31,000 N. It allowed the single-phase material to be obtained at the relative low temperatures of 900–1100 °C. To stabilize the structure, it was quenched from the synthesis temperature to room temperature. Desissa et al. [112] proposed a similar SSR procedure, although for starting materials, MgO, CoO, NiO, CuO, and ZnO oxides preliminarily obtained by a precipitation method were used. The optimal synthesis temperature was found to be equal to 950 °C. The ceramic samples with relative bulk density and Vickers hardness of 99.5% and 16 GP, respectively, were formed at 1200 °C for 15 h.

Chen at al. [113] obtained the (NiMgCuZnCo)O classical composition by a mechanochemical synthesis using only room temperature ball-milling without post-heat treatment. To use it as a catalyst, noble metal oxides were dispersed into the rock salt lattice by solid-state grinding assisted by HE stabilization (500 °C heat treatment for 2 h). The scheme of the method is shown in Figure 10a. Catalytic testing showed that for the 5 wt.% Ru-500 and 5 wt.% Pt-500 catalysis based on the HEO, the yields of CO were 45.7% and 46.1% with CO_2_ conversions of 45.4% and 47.8%, respectively.

Both 4- and 5-cationic systems, (CoMgNiZn)O and (CoCuMgNiZn)O were stabilized into a single rock salt structure directly using nebulized spray pyrolysis (NSP) in [114]. Schematically, this method is represented in Figure 10b [115]. Using flame spray pyrolysis (FSP) (shown in Figure 10c [116]) and reverse co-precipitation (RCP), the stabilization of the HEOs structure was reached only after the thermal treatment at 1000 °C. The application of these methods allowed nanocrystalline materials to be obtained. Wand et al. [117] proposed a wet-chemistry sol–gel strategy which was demonstrated to be useful for the synthesis of spherical mesoporous (NiCoCrFeMn)O of spinel structure with a high specific surface area up to 143 m^2^/g with a pore size of 5.5–8.3 nm and a unique spherical morphology (∼55 nm).

The importance of synthesizing rare earth oxide (REO)-based HEOs is driven by the technologically interesting properties of REOs and the wide range of their application, including those as solid-state electrolytes in SOFCs and other high-temperature electrochemical devices [118]. Djenadic et al. [107] suggested three “selection rules” as a guide to selecting the equiatomic compositions to produce single-phase REO-based HEOs: (1) cations should have similar ionic radii, (2) at least one of the constituent binary oxides should have a different crystal structure and the overall system should have distinctive electronegativity, and (3) at least one binary oxide pair should not have a complete miscibility gap at 0.5 mole fraction. They obtained a few REO-based HEOs using NSP technique. Nevertheless, according to the literature data, solid-state reaction and wet chemical methods are the most widely used for the synthesis of REO-based HEOs. New multicomponent equiatomic rare earth oxides containing 3–7 rare earth elements (Ce, Gd, La, Nd, Pr, and Sm) and Y in equiatomic proportions are synthesized using nebulized spray pyrolysis by Sarkar et al. [119]. All the systems crystallized as a phase pure fluorite type (Fm-3m) structure despite the high chemical complexity. Polycrystalline ceramics comprising various combinations of Ce, Gd, La, Nd, Pr, Sm, and Y oxides in equiatomic proportions were synthesized from the initial binary oxide powders in [120] using solid-state sintering in the air at 1500 °C for 10 h with a cooling rate of 3.3 °C/min. Slow cooling rates were applied to test the idea that the mechanism of phase stabilization in multicomponent REO systems is different from that of the high-entropy transition metal oxides. Single-phase HE compositions containing more than four elements with fluorite structure ((CeGdLaPr)O, (CeLaPrSm)O, (CeLaPrY)O), C-type structure ((CeGdLaPrY)O, (CeLaPrSmY)O) and B-type structure ((GdLaNdSm)O, (LaNdSmY)O, (GdLaSmY)O, and (GdLaNdSmY)O) were obtained. In contrast to the results of [107], the authors did not observe the stabilization effect of Ce^4+^, which they supposed may be related to the different temperature regimes used. A new transition metal/rare earth entropy-stabilized oxide with the composition of (Ce_0.2_Zr_0.2_Y_0.2_Gd_0.2_La_0.2_)O_1.7_ and possessing a fluorite structure was produced by Spiridigliozzi et al. [13] using a hydrothermal synthesis. The single-phase state was reached at calcination temperatures of 1100–1200 °C. The obtained powders of (CeZrYLaGd)O exhibited a good sinterability and allowed the sintering of ceramics at 1500 °C. A new, high-entropy lanthanide sesquioxide Gd_0.4_Tb_0.4_Dy_0.4_Ho_0.4_Er_0.4_O_3_ solid solution having a single phase, cubic-bixbyite structure with no phase transformation from room temperature to 1650 °C was synthesized by Tseng et al. [121] via a polymeric steric entrapment method, which has an advantage in fabricating homogeneous solid solutions.

A series of perovskite-like Sr(Zr_0.2_Sn_0.2_Ti_0.2_Hf_0.2_Mn_0.2_)O_3_, Sr(Zr_0.2_Sn_0.2_Ti_0.2_Hf_0.2_Nb_0.2_)O_3_, Ba(Zr_0.2_Sn_0.2_Ti_0.2_Hf_0.2_Ce_0.2_)O_3_, Ba(Zr_0.2_Sn_0.2_Ti_0.2_Hf_0.2_Y_0.2_)O_3-x_, Ba(Zr_0.2_Sn_0.2_Ti_0.2_Hf_0.2_Nb_0.2_)O_3_, and (Sr_0.5_Ba_0.5_)(Zr_0.2_Sn_0.2_Ti_0.2_Hf_0.2_Nb_0.2_)O_3_ HEOs was synthesized by Jiang et al. [122]. The authors used the SSR method; however, the initial commercial oxides were high-energy ball-milled for 6 h for better homogeneity. The milled powders were compacted and sintered at 1500 °C. Subsequently, most sintered specimens were cooled inside the furnace (power off), where the cooling rate was measured to be between 10 and 50 °C/min. A similar solid-state reaction method was successfully used by Gazda et al. [12] to obtain BaZr_1/7_Sn_1/7_Ti_1/7_Hf_1/7_Ce_1/7_Nb_1/7_Y_1/7_O_3−δ_ and BaZr_0.15_Sn_0.15_Ti_0.15_Hf_0.15_Ce_0.15_Nb_0.15_Y_0.1_O_3−δ_, which belong to a new class of high-entropy proton-conducting electrolytes for SOFCs. Recently, Teng et al. [123] synthesized (La_0.2_Nd_0.2_Sm_0.2_Gd_0.2_Yb_0.2_)_2_Zr_2_O_7_ nanopowder using combustion synthesis from a mixture of metal nitrates using glycine as an organic fuel. Transparent ceramic of the same composition was fabricated by Zhang et al. using combustion synthesized nanopowder [124]. Biesuz et al. [125] prepared high-entropy perovskite Sr((Zr_0.94_Y_0.06_)_0.2_Sn_0.2_Ti_0.2_Hf_0.2_Mn_0.2_)O_3−x_ from the oxide powder mixture using conventional sintering carried out by heating the specimens to 1400 °C and 1500 °C for 2 h with heating rate 2 °C/min up to 400 °C and 10 °C/min from 400 °C to the final sintering temperature) and by reactive spark plasma sintering (SPS) at 1450 °C and 1375 °C. Application of external pressure during SPS allowed the temperature to decrease. Schematically, the SPS method is represented in Figure 10e.

Different methods were developed to fabricate HEOs’ thin films. Sharma et al. [126] demonstrated an example of a single-crystal HE perovskite oxide by stabilizing Ba(Zr_0.2_Sn_0.2_Ti_0.2_Hf_0.2_Nb_0.2_)O_3_ in epitaxial thin film on SrTiO_3_ and MgO single-crystal substrates. (MgCo_x_NiCuZn)O [127] and (Mg_x_Ni_x_Co_x_Cu_x_Zn_x_Sc_x_)O films [128] films were synthesized using pulsed laser deposition. HEO-based mesoporous thin films with controlled thickness of (Cr_0.2_Mn_0.2_Co_0.2_Fe_0.2_Co_0.2_Ni_0.2_)_3_O_4_ was prepared by Einert et al. [129] on conductive fluorine-doped tin (FTO) substrates via the dip-coating technique. Liang et al. [130] managed to deposit Ba(Zr_0.2_Sn_0.2_Ti_0.2_Hf_0.2_Nb_0.2_)O_3_ film through the sol–gel method. Synthesis methods of HEC including HEOs have been summarized in a number of recent reviews [11,131,132].

Methods based on the use of HEOs obtained by the solid phase method or using various chemical methods may potentially be used for the preparation of thin films due to their simple organization and great prospects for scaling. Techniques such as inkjet printing or screen printing can be tried to improve the properties and morphology of HEO-based thin films. Screen and inkjet printing technology eliminate target-based evaporation without the use of high-tech equipment, which greatly reduces production costs. The use of such methods would lay the foundation for the use of HEO thin films in SOFCs, including large scale production. The method of electrophoresis, which makes it possible to use oxide materials of different dispersity for the film deposition [133] may also be promising for the deposition of SOFC functional layers based on HEOs. It should be noted that the electrochemical deposition method has been widely used for the deposition of HEAs [52].

### 3.3. SOFC-Related HEO Applications

Intensive research activity applied in the last decade to the synthesis and characterization of the functional properties of HEOs has, in many cases, allowed them to replace conventional oxide-based materials in different application areas (Figure 9b). Among the most important functional properties of SOFCs which should be considered together with high thermal and phase stability is that of electrical conductivity. As was noted in Section 3.1, predicting this property is very problematic and necessitates taking the defect structure of the material into consideration. The works on the advanced research on the electrical properties of HEOs were summarized in a recent review by Li et al. [134]. However, the authors provided a little information concerning the SOFC-related materials. The purpose of this section of the review was to consider recent works on the preparation of HEOs promising for the use in the SOFC functional layers such as solid electrolytes with different types of conductivity and electrodes (cathodes and anodes), their characterization and the SOFC performance when using these materials, topics which, until now, have been omitted in the existing reviews.

#### 3.3.1. HEOs’ Application as SOFC Electrolytes

A typical SOFC single cell comprises a porous cathode, a dense ceramic electrolyte and a porous ceramic–metal composite anode. The layers form a sandwich-type structure with a gas-tight electrolyte in the middle, which separates anode and cathode compartments. A fuel, supplied to the anode channel, undergoes an electrochemical oxidation reaction with emitting electrons that reach the cathode through an external circuit to enable participation in the electrochemical oxygen reduction reaction (ORR). In dependence on the electrolyte type, which can be oxygen ion-conducting or proton-conducting, oxygen ions or protons transfer to the other side of the fuel cell (anode for O^2−^ and cathode for H^+^) through its bulk to complete the final reaction (Figure 11) [135].

A solid-state electrolyte is a key part of a SOFC and it should meet some very important requirements:(1)Gas-tightness;(2)High level of ionic conductivity (0.1 S/cm at the operating conditions) and low ohmic resistance (0.2 Ω cm^2^);(3)Pure ionic conductivity both in oxidizing and reducing atmospheres;(4)High chemical stability over a wide range of oxygen partial pressures.

Electrical properties of the state-of-art solid-state electrolytes are presented in Figure 12. Yttrium-stabilized zirconia (YSZ), an oxygen ion-conducting electrolyte, is traditionally used in high temperature SOFCs and other electrochemical devices. It has acceptable ionic conductivity only at high temperatures (0.1 S/cm at 1000 °C). Therefore, fuel cells based on such electrolytes, even in a thin-film version, demonstrate high power, mainly in the high temperature range of 800–1000 °C [136]. Materials based on La_1-x_Sr_x_MnO_3-δ_ are traditionally used in YSZ-based cells due to excellent thermo-mechanical compatibility of these materials [137]. Under the condition of using doped CeO_2_-based buffer cathode layers preventing chemical interaction with YSZ [138], electrodes based on La_1-x_Sr_x_Co_y_Fe_1-y_O_3-δ_ (LSCF) [139], which are mixed oxygen ion and electron conductors (MIECs) possessing excellent catalytic activity to ORR, can also be successfully applied. As a fuel electrode, Ni cermets are mainly used [140]; however, oxide electrodes have been intensively developed recently [141,142].

A lot of efforts have recently been made to reduce the operating temperature of SOFC in order to increase their long-term stability by decreasing the degradation of materials, characteristics for SOFCs with high operating temperatures, as well as reduce the SOFC cost by complete replacement of noble metal electrodes with oxide ones and using ferritic stainless steel as an interconnector. However, due to the fact that SOFCs mainly comprise oxide materials, the conductivity of which, as a rule, is thermally activated, a decrease in temperature imposes special requirements on its level both for electrolyte materials [135,143] and electrodes [142,144,145,146].

The problem of maintaining SOFC efficiency at lower operating temperatures can be partially solved by searching for new highly conductive electrolytes. Bi_2_O_3_-based oxides with a conductivity level of approximately 0.1 S/cm at 600 °C [147,148] became a focus of attention for the application in SOFCs [149,150,151]. To date, stabilized Bi_2_O_3_ electrolytes are applied as the electrolyte in IT-SOFCs very seldom due to their poor chemical stability in reducing atmospheres, easy volatilization of Bi_2_O_3_ and, as a consequence, poor mechanical stability. The use of electrolytes based on Sc_2_O_3_-stabilized ZrO_2_ [152,153] and doped LaGaO_3_ [154], exhibiting high conductivity level in the intermediate temperature (IT) range opens up great prospects to their usage in IT-SOFC. However, their high cost along with the limited resources of scandium and gallium creates problems with the large-scale use of such electrolytes. In addition, the interaction of electrolytes based on ZrO_2_ with oxide cathodes, and LaGaO_3_-based electrolytes with cermet anodes requires the creation of buffer layers, which greatly complicates the cell design.

Application of highly conductive electrolytes based on doped CeO_2_ [143,155,156] allows reduction in the cell operating temperature down to 700 °C without loss of efficiency of the electrolyte. At 600 °C, the ionic conductivity of Gd-doped ceria can reach as high as 0.024 S/cm, which suggests that the thickness of the electrolyte layer must be no more than 10 μm in order to reach the target ohmic resistance of 0.2 Ω cm^2^. In comparison, the conductivity of YSZ is approximately 0.009 S/cm at this temperature. This means that the thickness of the YSZ electrolyte layer should be approximately 1 μm to reach the target ohmic resistance, which is practically difficult and costly to implement, especially in the large size cells [157]. The main drawback of doped ceria electrolytes is that Ce^4+^ → Ce^3+^ reduction under anode conditions causes the power efficiency loss due to a decrease in the open circuit voltage (OCV), and as a result, fuel efficiency, which can be as low as 80% at 600 °C. Thus, the operation of CeO_2_-based SOFCs should be shifted to lower temperatures. Many attempts were applied to enhance ionic conductivity such the electrolytes below 600 °C and extend their electrolytic region, particularly, applying multi-element doping [138].

Other promising oxygen ion-conducting electrolytes are oxide compounds with a 2D-layered structure. The best oxide ion conductivity of the Sr_1-x_A_x_Si_1-y_Ge_y_O_3-0.5(x+y)_ (A = Na or K) series was found in the composition Sr_0.55_Na_0.45_SiO_2.755_ (SNS) [158]. At 600 °C, the measured oxide ion conductivity of SNS approaches those of Bi_2_O_3_-based materials, and demonstrates its stability down to 10^−30^ atm. However, the search of compatible electrodes for these new electrolytes is required.

The use of proton-conducting electrolytes in IT-SOFCs is very promising since the main charge carrier, protons, have a low mass and size and thus high values of conductivity and low values of its activation energy in this temperature range [159,160]. The principle of operation of SOFC based on proton electrolytes is somewhat different from that for an oxygen ion electrolyte, as can be seen from the comparison provided in Figure 11b. In the case of H^+^-SOFC, the release of H_2_O occurs at the air electrode. Thus, in contrast to devices based on oxygen ion electrolytes, higher values of open circuit voltage can be achieved, since there is no effect of “dilution” of the fuel. This makes it possible to significantly increase the efficiency of cell operation in the SOFC mode compared with devices based on unipolar oxygen ion electrolytes and, especially, MIEC electrolytes based on CeO_2_. Another category of SOFCs, called a dual ion-conducting SOFC (D-SOFC), has attracted considerable attention recently. In this cell, the electrolyte permits simultaneous diffusion of both oxygen ions and protons. To enable the advantages of both O–SOFCs and P–SOFC Yang et al. [161] proposed a famous dual ion-conducting electrolyte BaZr_0.1_Ce_0.7_Y_0.1_Yb_0.1_O_3−δ_ (BZCYYb), which allows rapid transport of both protons and oxide ion vacancies. It achieved conductivity level of 0.013 S/cm at 500 °C. The dual ion conduction was also registered in doped CeO_2_-carbonate composites. The conductivity of the SDC/Na_2_CO_3_ nanocomposite electrolyte reached 0.02 S/cm in 5% H_2_ and 0.002 S/cm in air at 500 °C [162].

In the last few years, due to rapid growth of the number synthesized HEOs with different structures, which can be promising candidates for solid electrolyte applications, the electrical conductivity as the main functional property has become a research hotspot. Many attempts were applied to reach high-conductivity high-entropy fluorites. However, the results obtained in a number of studies indicate a reduced ionic conductivity of high-entropy oxide ceramics compared with the traditional electrolytes. Gild et al. [163] studied electrical properties of the single-phase HE compositions with a fluorite structure of (Hf_0.25_Zr_0.25_Ce_0.25_Y_0.25_)O_2-δ_ (HEFO1), (Hf_0.2_Zr_0.2_Ce_0.2_)(Y_0.2_Yb_0.2_)O_2-δ_ (HEF04B)_,_ (Hf_0.25_Zr_0.25_Ce_0.25_)(Y_0.125_Ca_0.125_)O_2-δ_ (HEFO5A), (Hf_0.25_Zr_0.25_Ce_0.25_)(Y_0.125_Gd_0.125_)O_2-δ_ (HEFO7A), and (Hf_0.2_Zr_0.2_Ce_0.2_)(Y_0.2_Gd_0.2_)O_2-δ_ (HEFO7B) successfully obtained via high-energy ball milling, spark plasma sintering, and annealing in air at 1500 °C with following furnace cooling with a rate of 50 °C/min. The relative density of sintered samples achieved over 95%, the EDX elemental maps verified compositional uniformity in the obtained samples. However, the electrical conductivities of HEOs, measured in dry air in the temperature range of 650–850 °C were significantly lower than that obtained for the YSZ reference sample. For example, at 850 °C, they changed in the range from 1.9 mS/cm (HEF04B) to 2.9 mS/cm (HEFO1), while for YSZ the conductivity value of 30 mS/cm was obtained. All the HEFOs examined possessed activation energies of 1.14–1.29 eV, while YSZ exhibited *Ea* equal to 1.0 eV. The authors argue that decreasing the conductivity may be related to a smaller gran size of HE ceramics as well as to a high doping level, exceeding the optimal value for the ZrO_2_-based ceramic equal to 8–10 mol.% [164]. Bonnet et al. [165] undertook extended research of three HEOs series to identify the factors causing a decrease in the ionic conductivity level in HEOs with a fluorite structure. They synthesized (Hf_0.2_Zr_0.2_Ce_0.2_)(Y_0.2_Gd_0.2_)O_2-δ_ composition, which was shown to exhibit the lowest *Ea* in the series of materials obtained in [163], by spark plasma sintering under reducing conditions with the following oxidization at 800 °C in air (HEFO7B#1) and through the co-precipitation method with final sintering at 1500 °C for 6 h with a heating ramp of 2.5 °C/min with following air quenching (HEFO7B#2). The conductivity values obtained in this study were slightly higher than that obtained in [163] (3.9 and 2.9 mS/cm at 800 °C for the HEFO7B#1 and HEFO7B#2, respectively). However, they were lower than that of YSZ (49 mS/cm at 800 °C), prepared for the comparison in this study as well. To discover the influence of the dopant concentration, the authors designed some new compositions with 3+ cation ratio of 15%, setting an oxygen vacancy amount at 3.5% of anionic sites, close to 4% for conventional YSZ [164]. To decrease additional lattice distortion with doping, Yb^3+^ was chosen as a dopant with an ionic radius of 0.0985 nm (coordination number equal 8) [166] close to “critical ionic radius”, calculated according to the empirical equation of Kim [167] as 0.0974 nm.

Despite of the minimization of $\delta r_{cat}$, the obtained single-phase composition, possessing fluorite structure, (Hf_0.327_Ce_0.456_Zr_0.217_)_0.85_Yb_0.15_O_1.93_ (labeled as Yb-15(1)), (Hf_0.163_Ce_0.442_Zr_0.395_)_0.85_Yb_0.15_O_1.93_ (labeled as Yb-15(2)), and (Hf_0_Ce_0.429_Zr_0.571_)_0.85_Yb_0.15_O_1.93_ (labeled as Yb-15(3)) showed approximately the same conductivity value as HEFO7B#2 (2.8 S/cm at 800 °C) (shown in Figure 13). The series of compounds (Hf_0.2_Zr_0.2_Ce_0.2_)_1-x_(Y_0.2_Gd_0.2_)_x_O_2-δ_ with x ranging from 0.25 to 0.60 (labeled as 7–25, 7–30, 7–40, 7–50, and 7–60), corresponding to $V_{O^{2-}}^{∙∙}$ concentration values from 6 to 15% was also prepared, considering that the cationic entropy $\Delta S_{cat}$ is almost constant; therefore, $\Delta S_{anion}$ provides the main contribution to an increase in $\Delta S_{conf}$ (shown in Figure 13). The conductivity measured for this series was approximately of the same level as for the previous one, the highest values, obtained for the sample with x = 0.25 equaled to 4.7 mS/cm at 800 °C. The conductivity results obtained in this study (at 600 °C) and the data for the conventional solid-state electrolytes in dependence on $\Delta S_{conf},\;\delta r_{cat}$, and $\Delta S_{anion}$ are shown in Figure 13. It is seen that the difference in the ionic conductivity values is not significant for the samples with $\Delta S_{conf}$ values in the range of 1.5–2.4 (Figure 13a), thus, this factor did not have the expected impact on its level. The authors concluded that ionic conductivity in HEOs can be inhibited due to lowering the oxygen mobility, caused by a local crystal lattice distortion due to relatively high $\;\delta r_{ion}$ values (Figure 13b). As a comparison, the data obtained for the multi-component electrolytes Gd_0.11_Y_0.04_Ce_0.40_Zr_0.45_O_1.925_ (labeled as 50YSZ/50GDC) [168] and Y_0.18_Ce_0.41_Zr_0.41_O_1.91_ (labeled as 0.18YSZ/Ce) [169] were presented in Figure 13, demonstrating a decrease in *σ_i_* for these multicomponent compounds compared with the conductivity values obtained for their parent compositions. Therefore, in the high-symmetry cubic fluorite structure based on ZrO_2_, introducing new element results in entropy stabilization, which is accompanied generally with larger atomic size differences, and as a result, with *σ_i_* degradation. Moreover, high deviation of the oxygen non-stoichiometry from the optimal value of 4% for ZrO_2_-based systems (Figure 13c) also can contribute to decreasing the *σ_i_* level due the processes of defects association, decreasing the number of free charge carriers, as well as migration energies.

The influence of Pr^4+^ on the electrical properties ZrO_2_-based compositionally complex oxides can be considered in the example of the (Zr_0.2_Hf_0.2_Pr_0.2_La_0.2_)Y_0.2_O_2-δ_, (Zr_0.1667_Hf_0.1667_Pr_0.1667_La_0.25_)Y_0.25_O_2-δ_, and (Zr_0.1429_Hf_0.1429_Pr_0.1429_La_0.2857_)Y_0.2857_O_2-δ_ solid solution (labeled as CCFO1, CCFO2, and CCFO3, respectively) of a fluorite structure obtained by Zhang at al. [14] using the SSR route. The final synthesis step was performed at 1500 °C for 4 h followed by cooling with a rate of 5 °C/min. The $\Delta S_{conf}$ values calculated for the CCFO1, CCFO2, and CCFO3 samples were 1.61 R, 1.58 R, and 1.55 R, respectively. The formation of the cubic fluorite structure was confirmed by Raman spectroscopy method, the obtained spectra were typical for ZrO_2_-based solid solutions with cubic structure (Figure 13b). The maximal conductivity value, registered for the CCFO2 sample as 5.35 × 10^−4^ S/cm at 600 °C, was the highest among the investigated ZrO_2_-based HEOs. The *Ea* value was as low as 0.66 eV in the temperature range of 300–750 °C. Considering that Pr^4+^ is easily reducible in air to Pr^3+^, and Ag electrodes used in the study for the conductivity measurements are not blocking, the measured conductivities can be a combination of ionic and electronic conductivity of *p*-type.

An exceptionally interesting group in the context of SOFC applications is CeO_2_-based rare earth multicomponent oxides. It is known, however, that for rare earth-doped ceria materials there is a certain threshold of the dopant content, above which the oxygen vacancy ordering occurs, reducing the symmetry to *Ia*-3 bixbyite (C-type structure) [170]. The cations in the fluorite structure are located at the center of the ideally perfect cube formed by eight coordinated oxygen ions, while in the C-type structure they are 6-fold coordinated and the cation on the 24*d*-site moves away from the center because of electrostatic repulsion with vacancies (Figure 13d). This affects significantly the distribution of interatomic distances, especially for M-M pairs. It results in decreasing the mobility of oxygen vacancies and ionic conductivity. This threshold is defined by the Re_2_O_3_ nature and usually reach 0.15–0.20 mol.% for the single-doped ceria [138]. Investigating lanthanide co-doping of ceria, van Herle et al. [172] established, that despite co-doped ceria with 3, 5, or 10 dopants exhibited *σ_i_* by 10–30% higher than that for the single-doped ceria with the same vacancy concentration, the conductivity threshold for such systems was even low, approximately 0.10–0.12 mol.%. The presence of multiple dopants in HEOs in the total amount of approximately 80% has significant impact the energy landscape of the lattice sites, as well as migration edge energies, which needs the design rules developed for the conventional doped ceria systems to be reconsidered.

Raman spectroscopy is a power tool to investigate the oxygen vacancies formation in the lattice (defect region 250 1/cm and 500–600 1/cm for doped ceria). Furthermore, the blue/red shift of Raman modes is directly linked to increase/decrease in the lattice distortions induced by the presence of larger/smaller cations in the HEO structure. Raman spectra of CeO_2_ has a main band at 464 1/cm which is due to the F_2g_ symmetric vibrational mode of the 8-fold Ce-O bond (CeO_8_) in fluorite structure (Figure 13e). For HEOs, the position of the main band exhibits a blue shift, which is caused by the chemical substitution which leads the compression of the crystal lattice, and intensity of the main band decreases. Intensity of the broad band at 570 1/cm, related to the presence of oxygen vacancies, increases, which can demonstrate a rapid increase in the oxygen vacancies amount with element number increase in ceria-based HEOs [119].

Single-phase high-entropy HEOs of the composition of CeGdDySmPrO_2-δ_, CeGdYSmPrO_2-δ_, and CeGdLaSmPrO_2-δ_ were obtained Yapryntsev et al. in [173] using a sol–gel method followed by compaction and free sintering at 1500 °C in air. The authors noted that the powdered materials were single-phase and possessed body-centered cubic structure (sp. gr. Ia-3) even at the step of calcination at 900 °C. The lattice parameters and samples’ structure were unchanged after the sintering. The conductivity of the CeGdLaSmPrO_2-δ_ ceramics reached 0.001 S/cm at 510 °C. The *Ea* value was even lower than that obtained in [14] and attained the values of 0.52–0.53 both in the low temperature (170–400 °C) and intermediate temperature (450–560 °C) regions. Dabrowa et al. [174] applied efforts to stabilize the fluorite structure in HEOs of the composition of CeGdNdSmPrO_2-δ_ and CeGdLaNdPrO_2-δ_ with the oxygen vacancy-ordered *Ia*-3 structure by Mo-doping. The single-phase (CeGdNdPrSmMo_0.5_)O_2-δ_ and (CeGdLaNdPrMo_0.5_)O_2-δ_ samples with fluorite structure were obtained by solid-state reaction method with final sintering for 36 h at 1500 °C with following furnace-cooling. Using Raman spectroscopy and XPS methods, a decrease in the vacancy content and dominating presence of Mo^6+^, Ce^4+^ and mixed Pr^3+/4+^ states was demonstrated. The measured conductivities CeGdNdSmPrO_2-δ_ and CeGdLaNdPrO_2-δ_ and those doped with Mo^6+^ in the amount of 0.5 were 2.3, 2.1, 1.0, and 1.3 mS/cm at 600 °C. The authors argued that doping with Mo^6+^, which stabilized the oxygen vacancy-disordered Fm-3m crystal structure, enabled the improvement of ionic conductivity. However, the Mo^6+^ doping resulted in decreasing the vacancy concentration and dilution of Pr^3+/4+^ states responsible for the electronic part of conductivity, thus the total conductivity values decreased. Several series of HEOs with a stabilized fluorite structure were obtained in a number of studies [119,120], which opens perspectives for further studies of their transport properties to reveal their potential for possible application as IT-SOFC electrolytes.

A series of highly dense high-entropy rare earth zirconates (Nd_0.2_Sm_0.2_Gd_0.2_Dy_0.2_Y_0.2_)_2_Zr_2_O_7_ (labeled as HEC-1), (Nd_0.2_Sm_0.2_Eu_0.2_Dy_0.2_Y_0.2_)_2_Zr_2_O_7_ (labeled as HEC-2), (Nd_0.2_Sm_0.2_Eu_0.2_Gd_0.2_Er_0.2_)_2_Zr_2_O_7_ (labeled as HEC-3), (Nd_0.2_Sm_0.2_Eu_0.2_Gd_0.2_Y_0.2_)_2_Zr_2_O_7_ (labeled as HEC-4), and (Nd_0.2_Sm_0.2_Eu_0.2_Gd_0.2_Dy_0.2_)_2_Zr_2_O_7_ (labeled as HEC-5) were prepared in a recent study of Wang et al. [171] by a high temperature solid-state reaction method with final sintering temperature of 1700 °C. It was demonstrated that by increasing the cation radius ratio of r(A^3+^)^/^r(B^4+^), a degree of order in high-entropy ceramics increased, and the ordered pyrochlore structure was maintained. Despite the high density of the obtained ceramics (95.7–98.8% relative density), the average grain size was relatively low (1.2–2.4 μm), which can be explained by sluggish diffusion characteristics of HEOs. The conductivity increased along with increasing the r(A^3+^)^/^r(B^4+^) ration, and the highest conductivity value of 7.1 × 10^−3^ S/cm (at 800 °C) was obtained for the HEC-5 sample, which was higher for the base low and medium zirconates (Figure 13f). The first principles calculations carried out for HEO pyrochlore systems using the Vienna ab initio simulation package (VASP) showed that higher ion conductivity is ensured through higher mobile vacancy concentration and lower activation energy. In pyrochlore structure, oxygen that jumps from 48*f*-site to 48*f*-site can form a continuous and energetically favorable pathway which can play a major role in the increase in ionic conductivity. Such a preferential pathway cannot be formed in a HEOs with the fluorite structure due to random occupation at the cation and anion sites.

Certain progress was achieved in the development of HEOs possessing proton conductivity. A series of high-entropy, single-phase perovskites of BaZr_0.2_Sn_0.2_Ti_0.2_Hf_0.2_Ce_0.2_O_3-δ_, BaZr_0.2_Sn_0.2_Ti_0.2_Hf_0.2_Y_0.2_O_3-δ_, BaZr_1/7_Sn_1/7_Ti_1/7_Hf_1/7_Ce_1/7_Nb_1/7_Y_1/7_O_3-δ_, and BaZr_0.15_Sn_0.15_Ti_0.15_Hf_0.15_Ce_0.15_Nb_0.15_Y_0.10_O_3-δ_ were synthesized by Gazda et al. [12] using a solid-state reaction method with final sintering temperatures in the range of 1400–1500 °C, dependent on composition. The idea of using high-entropy oxides as proton conductors was based on the possible influence of configurational entropy on the possibilities of obtaining high concentration and mobility of proton defects. BaZr_0.2_Sn_0.2_Ti_0.2_Hf_0.2_Y_0.2_O_3-δ_ exhibited the highest water uptake, the concentration of proton defects was 1.6 × 10^−2^ mol/mol, while for the rest of the obtained materials it was approximately by an order of magnitude lower. The total conductivity of the HEOs containing five elements on B-site was higher than that of the oxides containing seven. For all the obtained materials, total conductivity in humidified air was higher than that in dry air and total conductivity in the D_2_O-containing atmosphere was lower than in humidified air. The highest conductivity values in the series were obtained for the BaZr_0.2_Sn_0.2_Ti_0.2_Hf_0.2_Y_0.2_O_3-δ_ sample, at 600 °C: 4.4 × 10^−4^ S/cm in dry air and 6.0 × 10^−4^ S/cm in wet air. Both the conductivity values and the values of activation energy of conductivity were higher than those observed in low-entropy proton-conducting perovskites.

HEO of perovskite structure of BaSn_0.16_Zr_0.24_Ce_0.35_Y_0.1_Yb_0.1_Dy_0.05_O_3−δ_ (BSZCYYbD) was synthesized by Guo and He [175] using a sol–gel method with final sintering at 1500 °C. The protonic conductivity of BSZCYYbD was the highest ever reached in the high-entropy proton conductors and amounted 8.3 mS/cm at 600 °C. The conductivity of conventional Ba_1.03_Ce_0.6_Zr_0.2_Yb_0.2_O_3-δ_ and BaCe_0.6_Zr_0.3_Y_0.1_O_3-δ_ electrolytes are 6.1 and 7.4 mS/cm [176,177]. An anode-based cell with BSZCYYbD electrolyte (∼45 μm) and BaCo_0.4_Fe_0.4_Zr_0.1_Y_0.1_O_3-δ_ (BCFZY) cathode demonstrated a maximum power density of 318 mW/cm^2^ at 600 °C. The cell demonstrated sufficiently high OCV equal to 1.06 and 1.08 eV and 600 and 500 °C, respectively.

The literature data on the conductivity of HEOs are summarized in Figure 14.

The analysis of the data presented in Figure 14 shows that HEO electrolytes are most promising for the SOFCs operating below 600 °C due to lower activation energy compared to the conventional electrolytes. Further efforts should be applied to the search for new methods to stabilize fluorite HEOs without deterioration of their transport properties. HEOs with pyrochlore structure are highly stable under severe operation conditions, moreover, entropy disordering in such the oxides enhances their transport properties, which provides a point for their future development. HEO proton and dual conductors with perovskite structure are highly demanded due to their excellent conductivity, this provides grounds to expect growing interest in these materials in the coming years.

#### 3.3.2. HEOs’ Application as SOFC Cathodes

The main requirements for SOFC cathodes were formulated by Sun et al. [178] as follows:(1)Electrical conductivity of no less than 100 S/cm under the oxidizing conditions;(2)Thermal expansion compatible with electrolyte and interconnector materials;(3)Absence of chemical interactions between the electrolyte and interconnector materials;(4)Sufficient porosity (usually no less than 35%) for oxygen migration through the cathode to the cathode/electrolyte interface;(5)Phase stability under oxidizing conditions;(6)High catalytic activity for the oxygen reduction reaction (ORR);(7)Easy production and low cost.

The application of HEOs as prospective cathode materials for SOFCs is presented in a limited number of studies. However, HEOs were validated as cathodes both in SOFCs with traditional oxygen–ion-conducting electrolytes (Y-stabilized ZrO_2_ (YSZ), Gd-doped CeO_2_ (GDC), Sr, and Mg-doped LaGaO_3_ (LSGM)) as well as in a cell based on the proton-conducting electrolyte Zr and Y-doped BaCeO_3_ (BCZY). Among them, HEOs with a perovskite structure [17,179,180,181,182,183,184], double perovskite structure [185,186], K_2_NiF_4_-type structure [15], and with a spinel-like structure [187]. The literature data for the electrochemical performance for different types of SOFCs with HEO cathodes are summarized in Table 1.

Among HEOs with a perovskite structure, the cathode materials on the base of LaMnO_3_ (LMO) with five transition metal ions Mn, Fe, Co, Ni, and Cu incorporated in the B-site and five metal ions La, Pr, Nd, Sm, and Sr incorporated in the A-site should be noted accordingly: Sr-free LaMn_0.2_Fe_0.2_Co_0.2_Ni_0.2_Cu_0.2_O_3-δ_ (HE_LMO) [17] and La_0.2_Pr_0.2_Nd_0.2_Sm_0.2_Sr_0.2_MnO_3-δ_ (HE_LSM) [179].

LaMn_0.2_Fe_0.2_Co_0.2_Ni_0.2_Cu_0.2_O_3-δ_, prepared in [17] by a modified EDTA-citrate complexation process, possessed a hexagonal perovskite structure, all peaks on a XRD pattern (presented in Figure 15a) belonged to a *R-3cH* space group. The calculated $S_{conf}$ value for the HE_LMO was equal to 1.61 R, which was higher than the limit for HEOs ($S_{conf}\geq 1.5\;R)$. The results of the electrochemical testing in symmetrical cells showed [17] that in contact with YSZ electrolyte, the HE_LMO cathode demonstrated an enhanced performance compared with the base LMO cathode (ASR value of 0.212 and 0.532 Ω cm^2^ at 800 °C, respectively). The performance of the NiO-YSZ|YSZ|CGO|HE_LMO single cell at 800 °C (data in Table 1) was one and a half times higher than one for NiO-YSZ|YSZ|CGO|LMO. The stability test of the NiO-YSZ|YSZ|CGO|HE_LMO cell illustrated that the average output voltage of 0.75 V was observed at 700 °C for 50 h.

Even more improved power characteristics and increased long-term cell stability were obtained by Yang et al. upon the introduction of four cations into the A-site of the conventional L_a0.8_Sr_0.2_MnO_3-δ_ (LSM) cathode [179]. Equal-molar substitution of La in basic LMO with five metal cations resulted in the formation of La_0.2_Pr_0.2_Nd_0.2_Sm_0.2_Sr_0.2_MnO_3-δ_ (HE-LSM) and that caused a space group transformation from *R-3cH* for LMO to *Pnma* for HE_LSM. The lattice distortion of HE-LSM due to the uniform distribution of A-site cations had some advantages with an increased oxygen vacancies content, enhanced oxygen adsorption properties, and the formation of a disordered stress field around Sr, which diminished the migration of Sr cations. The NiO-YSZ|YSZ|GDC|HE_LSM anode-supported single cell with the thicknesses of the HE_LSM, GDC, and YSZ layers of 30, 5, and 10 μm, respectively (presented in Figure 15b), demonstrated the excellent power density values (Table 1, Figure 15c). The polarization resistance value of 0.40 Ω cm^2^ at 800 °C was measured for the HE_LSM cathode, and the long-term durability test of the single cell with the HE-LSM cathode illustrated 100 h stability of the average output voltage of 0.7 V at 700 °C. The data obtained in [179] showed that the use of the LMO-based HEOs as the cathodes is promising, both in terms of the enhanced power output and in suppressing the Sr cation segregation.

Shi et al. [180] systematically designed a series of LMO-based HEOs to investigate the effect of the A-site high-entropy composition on the structure and thermochemical/electrical properties. The authors revealed that the high-entropy effect proved to be superior to the conventional doping effect, which resulted in that fact that the obtained HEOs exhibited higher crystallographic symmetry with much higher cation size differences and at Goldschmidt tolerance factor values more highly deviated from 1. Among the studied La_0.2_Nd_0.2_Sm_0.2_Y_0.2_Gd_0.2_MnO_3-δ_ (HEALMO-1), La_0.2_Nd_0.2_Pr_0.2_Sr_0.2_Ba_0.2_MnO_3-δ_ (HEALMO-2), and La_0.2_Nd_0.2_Sm_0.2_Ca_0.2_Sr_0.2_MnO_3-δ_ (HEALMO-3) compositions, HEALMO-3 possessed excellent chemical compatibility with YSZ, as shown in Figure 15d, as well as the highest value of electrical conductivity in the series, which was slightly lower than the value for the conventional LSM at 800 °C (Figure 15e).

Among all types of perovskite oxides, significant interest is being devoted to cobaltites, that is, derivatives of LaCoO_3_. Such materials are used in solid oxide fuel cells as cathodes due to their excellent catalytic activity in the Oxygen Reduction Reaction (ORR). An original and efficient method for the synthesis of the perovskite-type HEOs of (Gd_0.2_Nd_0.2_La_0.2_Sm_0.2_Y_0.2_)CoO_3_, combining the coprecipitation hydrothermal method with an original approach to heat treatment, which comprises quenching utilizing liquid nitrogen as a cooling medium, was developed by Krawczyk at al. in [189]. As a result, a single-phase ceramic with high configuration entropy, crystallizing in an orthorhombic distorted structure was obtained. The authors discussed the possibility of fine-tuning of the semiconducting band gap via a subtle change in A-site stoichiometry. This study can be considered as a base for further development of the cobaltite-based materials for SOFC applications.

A single-phase perovskite La_0.2_Pr_0.2_Nd_0.2_Sm_0.2_Ba_0.1_Sr_0.1_Co_0.2_Fe_0.6_Ni_0.1_Cu_0.1_O_3-δ_ cathode material (HEP) with a cubic perovskite structure was prepared via a combustion method with annealing at the temperature of 1000 °C during 3 h in [181]. The calculated $S_{conf}$ of the obtained HEP was equal to 2.84 R. Conductivity and electrochemical measurements showed that HEP achieved a sufficient electrical conductivity level of 635.15 S/cm at 800 °C, which was higher than that for the well-established La_0.8_Sr_0.2_FeO_3-δ_ (LSF) cathode. For the symmetrical cell with the HEP-based cathode on the YSZ electrolyte substrate with GDC buffer layers deposited on both sides, the polarization resistance values were 1.39 and 0.31 Ω cm^2^ at 700 and 800 °C, respectively, while the corresponding polarization resistance values for the similar symmetrical cell with the LSF cathode were equal to 4.52 and 0.60 Ω cm^2^. The maximum power densities (MPDs) for the NiO-YSZ|YSZ|GDC|HEP anode-supporting cell were measured as 396.36 and 714.53 mW/cm^2^ at 700 and 800 °C, respectively. The authors presumed that HEP is a promising cathode material. Moreover, its electrochemical performance can be further optimized by using it in composites, as it was provided earlier, for example, for the composite cathodes on the base of LaNi_0.6_Fe_0.4_O_3-δ_ [149] and La_0.6_Sr_0.4_Co_0.2_Fe_0.8_O_3-δ_ [139,190,191,192,193].

Nanosized homogeneous powder materials of the La_1-x_Sr_x_Co_0.2_Cr_0.2_Fe_0.2_Mn_0.2_Ni_0.2_O_3-δ_ series (x = 0, 0.1, 0.2, 0.3, 0.4, and 0.5) were synthesized using a modified Pechini sol–gel method by Dabrowa et al. in [182]. With a strontium substitution increase in series, the thermal expansion coefficient decreased, and the electronic conductivity was raised along with an increase in the complex oxide stability to Cr contamination. In their following study [194], the same authors’ group showed that the Sr-free LnCo_0.2_Cr_0.2_Fe_0.2_Mn_0.2_Ni_0.2_O_3-δ_ (Ln = La, Pr, Nd, Sm, and Gd) series was characterized by a drop in electrical conductivity when decreasing the ionic radius of the lanthanide. Similar electrical behavior was observed for LnCo_0.2_Cr_0.2_Fe_0.2_Mn_0.2_Ni_0.2_O_3-δ_ in [195] (Ln = Sm, Eu, Gd) and in [196] (Ln = La, Nd).

La_0.7_Sr_0.3_Co_0.2_Cr_0.2_Fe_0.2_Mn_0.2_Ni_0.2_O_3-δ_ (HE_L7S3) composition, possessing a rhombohedral perovskite structure (sp. gr. *R-3c*), was chosen in [182] for the more detailed study due to its maximal conductivity value. The high $\;S_{conf}$ value equal to 1.61 R contributed to the structural stabilization of HE_L7S3. It should be noted that two-cationic equimolar compositions based on La_0.7_Sr_0.3_MnO_3_ (L7S3): La_0.7_Sr_0.3_Mn_0.5_Fe_0.5_O_3_ [197], La_0.7_Sr_0.3_Mn_0.5_Ni_0.5_O_3_ [198], La_0.7_Sr_0.3_Co_0.5_Ni_0.5_O_3_ [199], and La_0.7_Sr_0.3_Mn_0.5_Cr_0.5_O_3_ [200] were not obtained earlier as single-phase in contrast to the homogeneous rhombohedral La_0.7_Sr_0.3_Mn_0.5_Co_0.5_O_3_ [201] and La_0.7_Sr_0.3_Co_0.5_Fe_0.5_O_3_ [202] compositions. The maximum conductivity of HE_L7S3 reached 16.03 S/cm at 950 °C, and as is shown at Figure 15f, the average value of the linear coefficient of thermal expansion (LCTE) was equal to 16.0(3) × 10^−6^ 1/K at temperatures below 1000 °C [182]. The values of σ and LCTE for HE_L7S3, obtained in [182], were inferior from a practical point of view to the corresponding values equal to 251 S/cm at 900 °C and 13 × 10^−6^ 1/K for basic L7S3, reported for example in [203].

A button-type fuel cell with HE_L7S3 as a cathode was manufactured in [182]. The tested single cell was based on the La_0.8_Sr_0.2_Ga_0.8_Mg_0.2_O_3-δ_ (LSGM8080) electrolyte with a screen-printed GDC interlayer to avoid the chemical reactivity between the NiO-GDC anode and the LSGM electrolyte. The polarization resistance of the HE_L7S3 cathode with the LSGM8080 electrolyte was 0.126 Ω cm^2^ at 900 °C. The MPD value, presented in Table 1, for the NiO-GDC|GDC|LSGM8080|HE_L7S3 cell was remarkably lower than, for example, those obtained in [204] for the button-type SNMM|LSGM8882|SDC|LSCF cell (625 mW/cm^2^ at 850 °C) with the conventional La_0.7_Sr_0.3_Co_0.9_Fe_0.1_O_3-δ_ (LSCF) cathode, the La_0.88_Sr_0.12_Ga_0.82_Mg_0.18_O_3-δ_ (LSGM8882) electrolyte, the Sr_2_NiMoO_6-δ_ (SNMM) anode, and with the Ce_0.8_Sm_0.2_O_2-δ_ (SDC) interlayer between the LSCF cathode and the LSGM8882 electrolyte. It is worth noting, however, that the use of the HEO cathode on the LaGaO_3_-based electrolyte does not require the use of a cathode buffer layer, which simplifies the cell fabrication. Thus, the development of HEO cathodes for the LaGaO_3_-based electrolyte cells with oxide anodes, stable against the chemical interaction with this kind of the electrolyte, is seen as a positive direction for further studies.

It was shown in [183] that HEO on the LSCF-base, La_0.2_Sr_0.2_Pr_0.2_Y_0.2_Ba_0.2_Co_0.2_Fe_0.8_O_3-δ_ (LSPYB), was characterized by excellent Cr-tolerance, demonstrating a level of degradation of the NiO-YSZ|YSZ|GDC|LSPYB cell of nearly 0.25% 1/kh during a 41-day test, while degradation of the similar cell with the LSCF cathode in the same condition showed a 100% increase during the first day. The MPD value, obtained in this study for the cell with the LSPYB cathode (Table 1) was the highest among those obtained for the cells with YSZ electrolyte with GDC buffer layer.

In summarizing the data on HEO with a perovskite structure, it can be concluded that multiple rare earth and alkaline earth substitutions at the A-site provides the most promising results. Optimal selection of the anode material for the anode and electrolyte-supporting cells with HEO cathodes can influence positively the entire cell performance and should be considered in the future.

As a representative of another prospective class of the SOFC cathode materials, double perovskites, HEO of SmBa(Mn_0.2_Fe_0.2_Co_0.2_Ni_0.2_Cu_0.2_)_2_O_5+δ_ composition (HE-SBC) with a double perovskite structure was for the first time obtained by Ling et al. in [185] using a modified Pechini method. The electrochemical activity of HE-SBC was enhanced by the addition of GDC with a molar ratio of 7:3, forming a HE-SBC-GDC composite cathode. The *R_p_* values of symmetrical cells with HE-SBC and HE-SBC-GDC cathodes on the YSZ substrate were 1.04 and 0.09 Ω cm^2^ at 800 °C, respectively. Considering the obtained MPD values for the NiO-YSZ|YSZ|GDC|HE-SBC and NiO-YSZ|YSZ|GDC|HE-SBC-GDC anode-supported single cells (Table 1), the authors of [185] suggested that the improved performance of the composite cathode was due to the enlarged three-phase interface.

The high-entropy oxide (La_0.2_Pr_0.2_Nd_0.2_Sm_0.2_Gd_0.2_)_2_CuO_4_ (HE_LCO) with a layered structure was synthesized by Shijie et al. in [15] using a sol–gel method. The obtained data showed that HE_LCO crystallized in a tetragonal structure (sp. gr. *I4/mmm*), the TEC value for HE_LCO in the temperature range of 100–800 °C was 12.19 × 10^−6^ 1/K, which is similar to values for layered Ruddlesden-Popper phases of the first order [205,206]. The ASR value for the HE_LCO cathode on the CGO substrate obtained at 700 °C was equal to 0.52 Ω cm^2^, which was remarkably lower than ASR values reported for the same temperature for HEOs in [179,181,182], and for the phases with Ruddlesden-Popper structure, for example, La_0.4_Sr_1.6_NiO_4_ (*R_p_* = 7.9 Ω cm^2^) in [207], La_1.7_Sr_0.3_NiO_4_ (*R_p_* = 12.8 Ω cm^2^) in [208], La_1.7_Ca_0.3_NiO_4_ (*R_p_* = 4.01 Ω cm^2^) in [209], La_1.4_Sr_0.6_Ni_0.5_Fe_0.5_O_4+δ_ (*R_p_* = 15.8 Ω cm^2^) in [210], and Cu-doped Nd_1.6_Ca_0.4_NiO_4_ (*R_p_* = 2.73–4.79 Ω cm^2^) in [206]. However, it was higher than the ASR value of 0.3 Ω cm^2^ for La_2_CuO_4+δ_, measured in [211]. Successful preparation of (La_0.2_Nd_0.2_Gd_0.2_Tb_0.2_Dy_0.2_)_2_CuO_4_ and (La_0.2_Pr_0.2_Nd_0.2_Sm_0.2_Eu_0.2_)_2_CuO_4_ HEOs with Ruddlesden-Popper structure, similar to HE_LCO, was reported in [212].

The maximum power density value of the anode-supported NiO-YSZ|YSZ|CGO|HE_LCO fuel cell obtained in [15] is presented in Table 1. Moreover, the testing of the long-term stability for the NiO-YSZ|YSZ|CGO|HE_LCO cell with 0.5 V load revealed the cell performance stability for 70 h at 700 °C. If the MPD value of the HE-LCO cell is compared with the literature data on the cell performance with another A_2_BO_4_-type oxides as cathode materials, it is worth noting that the MPD values of analogues cells depend strongly on the chosen type of SOFCs construction and their materials. For example, for the electrolyte-supporting SNMM|LSGM8882|SDC|LNO cell with the La_2_NiO_4+δ_ (LNO) cathode, for the anode-supporting Ni-YSZ|YSZ|SDC|LSNO cell with the La_1.75_Sr_0.25_NiO_4+δ_ (LSNO) cathode, and for the Ni-BCZY|BCZY|LNZO-BZY cell with the La_2_Ni_0.75_Zn_0.25_O_4+δ_-BaZr_0.8_Y_0.2_O_3-δ_ (LNZO-BZY) composite cathode and proton-conducting electrolyte BaCe_0.7_Zr_0.1_Y_0.2_O_3-δ_ (BCZY), the maximum power densities of 134, 269, and 1070 mW/cm^2^ at 700 °C were obtained, respectively [213,214,215]. When considering high-entropy stabilization of materials with layered structure, it should be noted that there are a few studies concerning the synthesis and featured SOFC applications of high-entropy layered nickelates, which are considered as highly relevant cathode materials for SOFCs with both oxygen ionic [216,217,218], and proton-conducting [219,220,221] electrolytes.

In a solid oxide fuel cell with a proton-conducting BaCe_0.7_Zr_0.1_Y_0.2_O_3−δ_ (BCZY) electrolyte (H^+^-SOFC), the spinel HEO Fe_0.6_Mn_0.6_Co_0.6_Ni_0.6_Cr_0.6_O_4_ (FMCNC) was used as a cathode in a single cell for the first time in [187]. The FMCNC powder, prepared via a modified sol–gel method, was deposited on the electrolyte layer followed by the NiO-BCZY anode (Figure 16a). With the aim of revealing the influence of the HEO cathode on the performance of the single cell, the cell with the Mn_1.6_Cu_1.4_O_4_ (MCO) cathode was constructed and examined under the same conditions. The obtained results for the NiO-BCZY|BCZY|FMCNC fuel cell (Figure 16b, Table 1) demonstrate the excellent performance of the cell with HEO cathode: the cell with the MCO cathode exhibited a maximum power density of 593 mW/cm^2^ under the same experimental conditions. The study of the long-term stability at 600 °C for the cell with the FMCNC cathode showed its stability for 100 h. In addition, the ASR value of 0.057 Ω cm^2^ at 700 °C for the FMCNC cathode was the lowest value for the H^+^-SOFCs reported in [221,222,223,224,225,226], and the linear thermal expansion coefficient for FMCNC was estimated in [83] as 9.7(2) × 10^−6^ 1/K.

The study by the first-principles method provided in [187], proved that the FMCNC high-entropy oxide possessed the best catalytic activity towards ORR in comparison with individual oxides, benefiting the cell performance. The mechanism of the FMCNC cathode high performance is founded on the high-entropy structure of the FMCNC oxide, that has both reduced hydration energy and an O *p*-band center closer to the Fermi level. With these properties, the high-entropy oxide improves the protonation and oxygen reduction reaction (ORR) activity for the cathode.

Also, spinel-type HEOs of FMCNC in [228,229,230,231]; Co_0.6_Cr_0.6_Fe_0.6_Mg_0.6_Mn_0.6_O_4_ and Cr_0.6_Fe_0.6_Mg_0.6_Mn_0.6_Ni_0.6_O_4_ in [83]; Al_0.6_Cr_0.6_Fe_0.6_Mn_0.6_Ni_0.6_O_4_, Co_0.6_Cr_0.6_Al_0.6_Mn_0.6_Ni_0.6_O_4_, Co_0.6_Cr_0.6_Fe_0.6_Al_0.6_Ni_0.6_O_4_, and Co_0.6_Cr_0.6_Fe_0.6_Mn_0.6_Al_0.6_O_4_ in [232]; Ni_0.6_Co_0.6_Mn_0.6_Fe_0.6_Ti_0.6_O_4_ in [233]; Mg_0.6_Ti_0.6_Zn_0.6_Cu_0.6_Fe_0.6_O_4_ in [85]; and Co_0.6_Mn_0.6_Fe_0.6_Zn_0.6_Ti_0.6_O_4_ in [234] were successfully prepared, demonstrating excellent structural stability. Authors of [234] explained this fact by the disordered intercalation of multivalent cations in two spinel sublattices and the enhanced formation of oxygen vacancies, which concurs with the conclusions of work [235] based on energy dispersive spectra and mapping about random, uniform, and equiatomic distribution of all cations between sublattices. However, Sarkar et. al. in [236], considering FMCNC, established this in cation distribution that deviated from the generally accepted theory of entropy-driven disordering of cations in HEOs for the first time.

Oxide systems with $S_{conf}$ value in the range of $1.0\;R\;\leq S_{conf}<1.5\;R$, considered as medium-entropy oxides, were investigated as cathode material for SOFCs in [188]. The SrFe_0.25_Ti_0.25_Co_0.25_Mn_0.25_O_3-δ_ (SFTCM25) perovskite, prepared in [188] via a citric acid nitrate combustion method, possessed a cubic perovskite structure (sp. gr. *Pm-3m*) after calcining at 950 °C, such as SrTi_0.2_Y_0.2_Zr_0.2_Sn_0.2_Hf_0.2_O_3_, reactive-flash sintered at 1000 °C in [237]. This was in contrast to the data on the Ba_0.2_Mg_0.2_Ca_0.2_Sr_0.2_Pb_0.2_TiO_3_ perovskite synthesized in [238] using a wet chemical method, annealed at 1200 °C, and the Bi_0.2_Na_0.2_Ba_0.2_Sr_0.2_Ca_0.2_TiO_3_, Bi_0.2_Li_0.2_Ba_0.2_Sr_0.2_Pb_0.2_TiO_3_, Bi_0.2_Na_0.2_Ba_0.2_Sr_0.2_Pb_0.2_TiO_3_, Bi_0.2_K_0.2_Ba_0.2_Sr_0.2_Pb_0.2_TiO_3_, and Bi_0.2_Ag_0.2_Ba_0.2_Sr_0.2_Pb_0.2_TiO_3_ perovskites, prepared in [239], all possessed a tetragonal structure.

The TEC value for the SFTCM25 ceramics, annealed in air for 4 h at 1200 and 1100 °C, was equal to 16.96 × 10^−6^ 1/K at 50–900 °C, [188]. The electrical conductivity value of SFTCM25 was measured as 11.4 S/cm at 600 °C, which is below the required conductivity level for the cathode. In addition, the obtained conductivity of medium-entropy perovskite SFTCM was lower compared with those for both conventional and high-entropy cathode materials (for example, LSM and HEALMO-3 in [180], and LSF and HEP in [181]). The value of SFTCM25 polarization resistance for the different elementary reactions of the ORR reaction (presented at Figure 16c), measured for the symmetrical cell on the LSGM8282 substrate, was equal to 0.147 Ω cm^2^ at 700 °C. It should be noted that the *R_p_* values for SFTCM25 were higher than ones for perovskites with a lower value of $S_{conf}$. Lower polarization resistance values were obtained for SrFe_0.5_Ti_0.2_Co_0.2_Mn_0.1_O_3-δ_ (SFTCM5221) in [188], and for SrCo_0.5_Fe_0.2_Ti_0.1_Ta_0.1_Nb_0.1_O_3-δ_ in [240], equal to 0.072 and 0.033 Ω cm^2^ at 700 °C, respectively. The polarization resistance data correlate well with oxygen surface exchange coefficients (*K_ex_*) and oxygen bulk diffusion coefficients (*D_chem_*), determined in [188] at 700 °C for SFTCM25 (6.155 × 10^−4^ cm/s and 13.30 × 10^−5^ cm^2^/s, respectively) and for SFTCM5221 (7.818 × 10^−4^ cm/s and 8.234 × 10^−5^ cm^2^/s, respectively). The above-mentioned correlation means that the perovskite’s catalytic activity decreases as the configuration entropy increases. However, it should be noted that the SFTCM25 cathode polarization resistance on the LSGM8282 electrolyte investigated at 700 °C in the synthetic air atmosphere (20 vol.% O_2_/80 vol.% N_2_) was stable for more than 100 h. Based on the fact that SFTCM25 perovskite showed a decay rate ten times lower in comparison with SFTCM5221, as it is shown in Figure 16d, the authors concluded that high entropy may sufficiently improve cathode stability due to the Sr segregation.

The MPD values of the electrolyte-supported single cell NiO-CGO|LDC|LSGM8282|SFTCM25 with LSGM layer with a thickness of 250 μm and a La_0.4_Ce_0.6_O_2-δ_ (LDC) buffer layer were 450, 850, and 1020 mW/cm^2^ at 700, 800, and 850 °C, respectively [188]. In addition, the single cell with the SFTCM25 cathode with a current load of 400 mA/cm^2^ demonstrated stability at 700 °C for 70 h. Obtained data allowed the authors to make a conclusion about the promising prospects of the medium-entropy perovskite SFTCM25 application in IT-SOFCs.

Another method of constructing a stable high-entropy oxide system was described in [227]. Baiutti et al. demonstrated entropy stabilization in the interface region of the LSM-SDC nanocomposite material, organized in the form of a vertically aligned nanostructure (VAN), which increased the temperature stability of the potential electrode material and accelerated the ORR reaction. The VAN structure, designed by the authors [227], consisted of LSM columnar regions with a thickness of 5 nm and a length of 100 nm, oriented across the plane of the YSZ substrate and located in the main SDC matrix, as it is shown schematically in Figure 16e. The formation of the VAN structure was carried out via a pulsed laser deposition of LSM and SDC mixture (1:1 wt.%) on the YSZ substrate, followed by annealing at 1300 °C for 4 h. According to the EDS analysis, there was no clear separation between the LSM and SDC regions and a certain degree of mixing of the La, Sm, and Ce cations (Figure 16f) was observed, that can correspond to the formation of La_0.75_Sr_0.125_Sm_0.125_MnO_3_ and La_0.625_Sr_0.125_Ce_0.25_MnO_3_.

The long-term polarization resistance measurements, provided in [227] for the electrolyte-supporting Ag|YSZ|Au-VAN cell, showed that the LSM-SDC nanocomposite, formed as a VAN structure, was stable during ageing at a temperature of 700 °C for 100 h in contrast to the conventional LSM, which degraded easily, as was shown in [179]. The authors [227] explained that the temperature stability of the VAN structure was caused by a local effect of entropy stabilization rising in the LSM-SDC boundary region owing to Sr-diffusion suppress. They considered that application of the VAN architecture would be a significant step forward in the energy conversion devices’ design in future.

#### 3.3.3. HEOs’ Application as SOFC Anodes

Requirements for SOFC anode material were summarized in [241] as follows:(1)Large surface area of triple phase boundary for maximizing the anodic reactions;(2)Enhanced porosity (approximately 50% for supporting anode layer and 35% for functional anode layer) facilitating a gas transport;(3)Stability in reducing atmospheres;(4)High electronic conductivity;(5)Thermal compatibility with other construction materials of the cell;(6)High electro-catalytic activity for hydrogen oxidation reaction.

Design of alternative materials for carbon- and sulfur-tolerant anodes in IT-SOFCs is currently a significant problem. The development of long-term stable anodes on the base of complex oxides with double perovskite structure Sr_2_FeMoO_6_ [141,242,243,244,245] and Sr_2_VMoO_6_ [246] for LaGaO_3_-based electrolytes, possessing the highest conductivity in the intermediate temperature range [154] is one of the possible solutions to this problem. The authors of work [18] suggested combining the high electronic conductivity of Sr_2_VmoO_6_ (SVMO) and high catalytic activity and sufficient ionic conductivity of Sr_2_FeMoO_6_ (SFMO) and prepared a single medium-entropy perovskite SrV_1/3_Fe_1/3_Mo_1/3_O_3_ (SVFMO) with V, Fe, and Mo cations in the *B*-position. The calculated value of $S_{conf}$ for SVFMO was equal to 1.1 R, which corresponded to medium entropy and was higher than the values for SFMO and SVMO (0.69 R).

The SVFMO powder was synthesized [18] using solid-state reaction with the final calcination at 1200 °C in atmosphere of 5% H_2_/Ar for 10 h. It was established that SVFMO conductivity was 508.4–207 S/cm in temperature range of 30–800 °C, which was higher than that for SFMO, measured for comparison, with Sr_2_NiMoO_6_ [247,248], Sr_2_MgMoO_6_ [249,250], and Sr_2_Ni_1-x_Mg_x_MoO_6_ [251,252,253,254,255], but it was lower than the conductivity of SVMO [18,246]. For this reason, the conductivities of porous SVMO, SVFMO, and SFMO samples were 78.8, 70.,1 and 37.9 S/cm at 800 °C, respectively. The average TEC value for SVFMO was measured as 17.7 × 10^−6^ 1/K in the range of temperatures of 30−800 °C, which ensured its satisfactory compatibility with conventional electrolyte materials YSZ, LSGM, and SDC [256] due to thermal expansion limitations [257]. The authors concluded that the high value of conductivity and optimal value of CTE for SVFMO met the requirements for SOFC anode layers.

EIS studies of the symmetrical cells of the anode|SDC|LSGM8282|SDC|anode design with the developed SVMO, SFMO, and SVFMO anodes demonstrated the *R_p_* values at 850 °C equal to 0.825, 0.522, and 0.371 Ω cm^2^, respectively, [18]. The lower *R_p_* for SVFMO was explained by the higher catalytic activity and faster charge carrier transport due to an increase in oxygen vacancies and small polaron couples. The electrolyte-supported anode|LSGM8282|LSCF single cell with the LSGM8282 electrolyte thickness of 250 μm was manufactured and tested at temperatures of 750–850 °C. The MPD values for the cell with the SVMO anode were equal to 55, 135, and 239 W/cm^2^ at 750, 800, and 850 °C, respectively. The MPD values for the cell with the SVFMO anode reached 267, 484, and 720 W/cm^2^ at 750, 800, and 850 °C, while the MPD values for the cell with the SFMO anode were lower and amounted to 259, 389, and 630 W/cm^2^ at 750, 800, and 850 °C, respectively. The obtained power density for the cells with the SVMO and SFMO anodes correlate sufficiently with MPD values for single cells with double perovskite anodes and LSGM electrolyte, known in the literature [204,213,248,252,258]. Therefore, the authors of [18] concluded that medium-entropy perovskite SrV_1/3_Fe_1/3_Mo_1/3_O_3_ demonstrated high conductivity, a medium range TEC value, and enhanced performance under SOFC operating conditions may be considered as a prospective anode material for IT-SOFCs.

## 4. Conclusions

The present review is focused on the application of high-entropy alloys (HEAs) and high-entropy oxides (HEOs) in the technology of solid oxide fuel cells (SOFC). In addition, the theoretical conditions for the formation of high-entropy materials (HEMs) and methods for the synthesis of powders, the fabrication of bulk ceramics, and films based on HEMs were also presented. It was shown that, HEAs, owing to their beneficial properties of high chemical resistance, sluggish diffusion of components, and their adequate electrical conductivity are considered to be suitable materials for obtaining diffusion barrier layers on metal interconnectors in SOFCs. HEOs with proton conductivity have proven to be very promising electrolytes for application in SOFCs operating at temperatures below 600 °C. It was revealed that the most prospective SOFC application of HEOs with perovskite structure was the development of cathode materials with enhanced catalytic activity and reduced polarization resistance due to suppressed Sr-segregation. Furthermore, the creation of vertically oriented heterostructures with local mixing of components in the interface region demonstrated an improvement in the stability of the cathode material due to the entropy factor. Medium-entropy oxides were considered as anodes in SOFCs with internal reforming, utilizing hydrocarbon fuel due to their high stability in reducing atmospheres. It was ultimately concluded that by utilizing the methods of atomistic modeling, computational thermodynamics, artificial intelligence previously developed for HEAs, and model approaches for the prediction of the HEOs’ properties, a strong base would be established for the design of new entropy-stabilized SOFC materials with optimized functional properties.

This overview of the recent achievements in the development of new HEMs will be used as a guide in our planned experimental studies in this field. It is hoped that it will also be of interest for other specialists in SOFCs and other electrochemical applications. We intend to develop a method to synthesize high-entropy ceramic materials in the form of thin films to create SOFC functional layers in order to increase efficiency and improve long-term stability.

## Figures and Tables

**Figure 1 materials-15-08783-f001:**
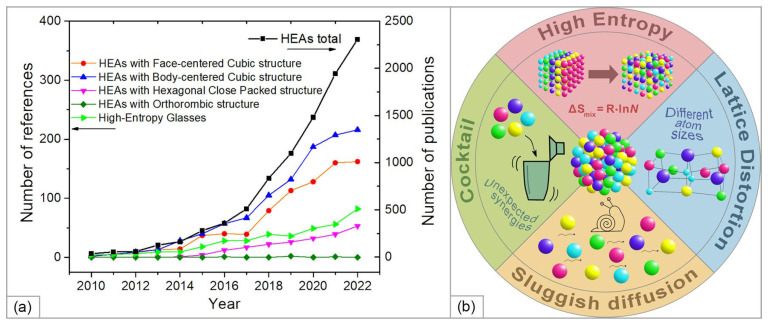
(**a**) Recent publications on the development and application of high-entropy alloys (HEAs) of different structures (for 2010–2022 years, according to the Scopus data); (**b**) schematic illustration of the properties and characteristics of HEAs.

**Figure 2 materials-15-08783-f002:**
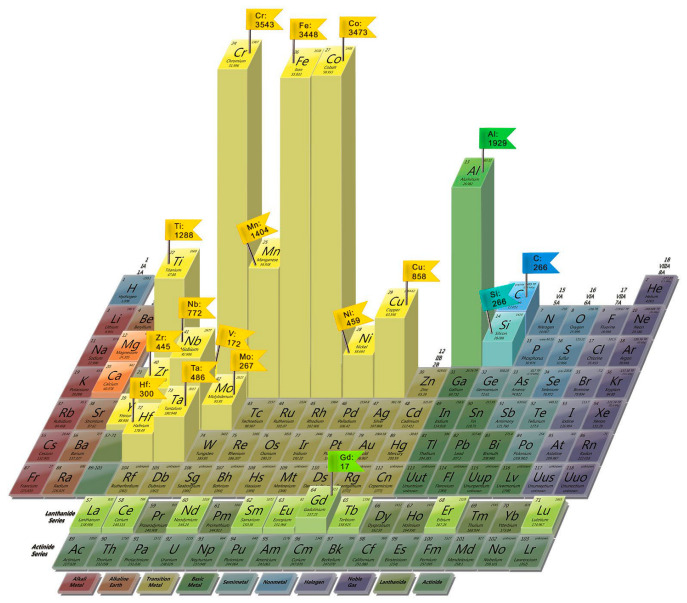
Frequency of use of elements in HEAs, Scopus data from years 2010–2022 (shown as vertical lines with corresponding numbers, elements used less than 10 times shown without numbers).

**Figure 3 materials-15-08783-f003:**
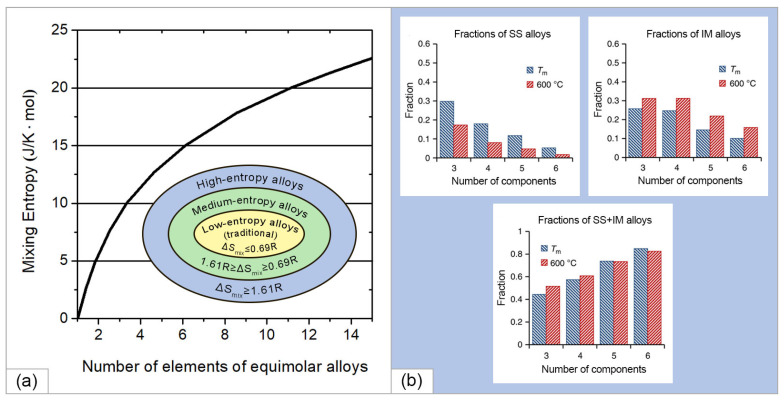
(**a**) The ratio between the entropy of mixing and the number of elements in a completely disordered equimolar alloy and inset in (**a**) categories of alloys based on the entropy approach (constructed using the data presented in [30] with permission of the LAVOISIER); (**b**) calculated fraction of different type of alloys on the number of components at the melting temperature (*T*_m_) and at 600 °C (reproduced from [31]).

**Figure 4 materials-15-08783-f004:**
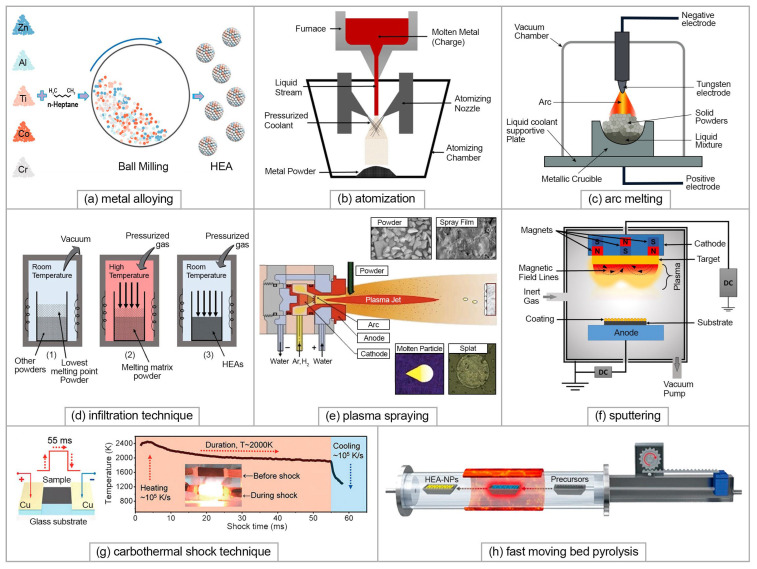
HEA synthesis methods: (**a**) metal alloying; (**b**) atomization; (**c**) arc melting; (**d**) infiltration; (**e**) plasma spraying; (**f**) sputtering; (**g**) carbothermal shock technique; (**h**) fast-moving bed pyrolysis ((**a**,**g**,**h**) are reprinted with permission from [28], Copyright (2020) American Chemical Society; (**b**,**d**,**f**) are reproduced from [50] with permission of Springer Nature; (**e**) is reproduced from [51] with permission of the Royal Society of Chemistry).

**Figure 5 materials-15-08783-f005:**
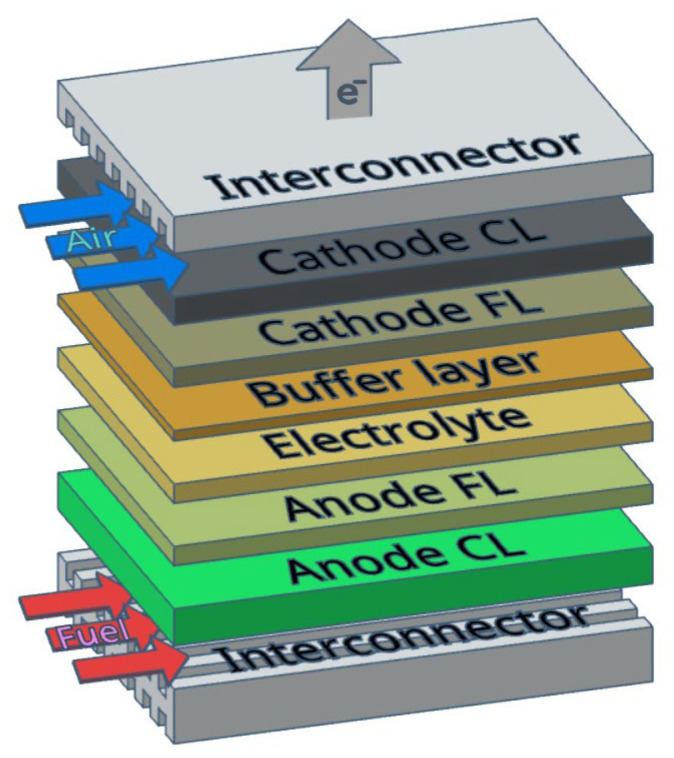
The typical components of a planar solid oxide fuel cell.

**Figure 6 materials-15-08783-f006:**
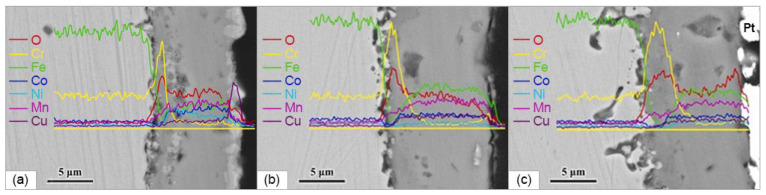
Cross-sectional images of the FeCoNiMnCu (1:1:1:1:1:1) HEA coating on SUS 430 steel after thermal treatment at 800 °C in air: (**a**) 1 week; (**b**) 5 week; (**c**) 10 weeks (reproduced from [19] with the permission of Elsevier).

**Figure 7 materials-15-08783-f007:**
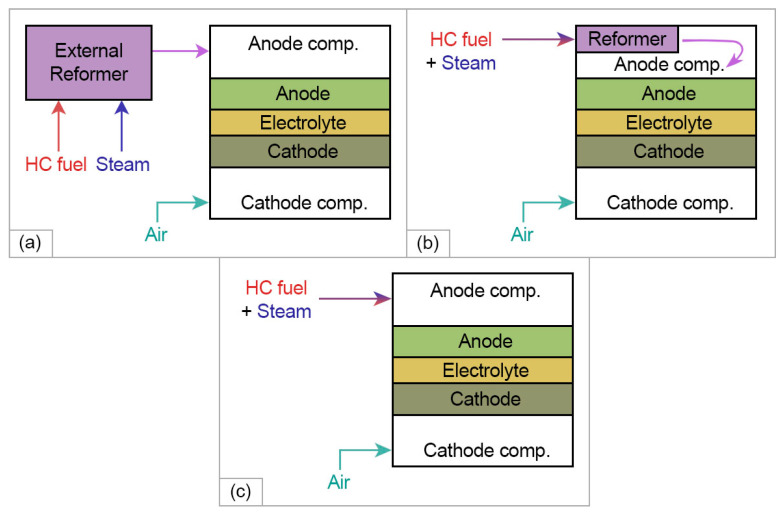
Schematic representation of (**a**) external reforming SOFC, (**b**) indirect internal reforming SOFC, and (**c**) direct internal reforming in SOFCs.

**Figure 8 materials-15-08783-f008:**
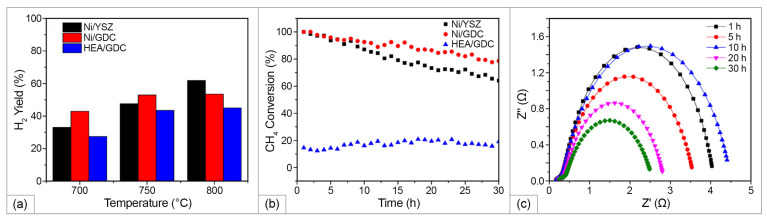
Methane steam reforming using Ni/YSZ, Ni/GDC, and HEA/GDC anode catalysts: (**a**) yield of hydrogen at reaction temperatures of 700, 750, and 800 °C, 1 atm, S/C = 2 and GHSV = 45,000 1/h; (**b**) time-on-stream stability test at 600 °C and S/C = 1 for 30 h; (**c**) time dependence of the overall ohmic and non-ohmic resistance of the HEA/GDC-Ni/ScSZ|ScSZ|LSM/YSZ cell at 750 °C (reproduced from [95]).

**Figure 9 materials-15-08783-f009:**
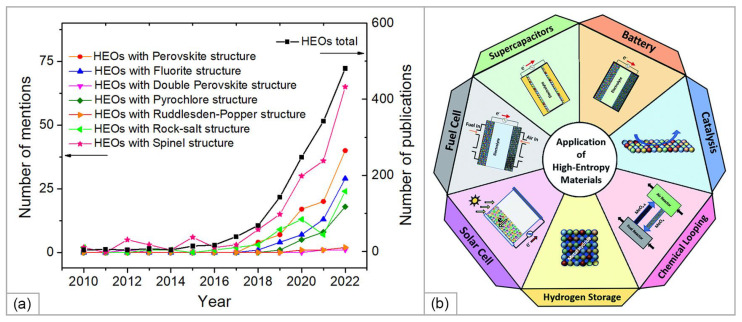
(**a**) Recent publications (on the development and application of high-entropy oxides (HEOs) of different structures (years 2010–2022, according to Scopus data); (**b**) schematic illustration of advanced applications of high-entropy ceramic materials including HEOs (reproduced from [103] with permission of the Royal Chemical Society).

**Figure 10 materials-15-08783-f010:**
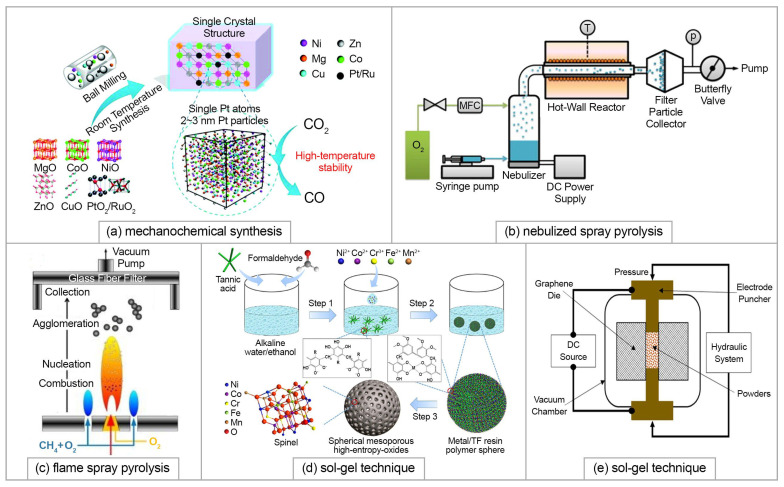
Various HEOs synthesis techniques: (**a**) mechanochemical synthesis; (**b**) nebulized spray pyrolysis; (**c**) flame spray pyrolysis; (**d**) sol–gel technique; (**e**) spark plasma sintering ((**a**) is reprinted with permission from [113], Copyright (2019) American Chemical Society; (b) is adopted from [115] with permission of Elsevier; (c) is adopted from [116] with permission of Royal Society of Chemistry; (d) is reprinted with permission from [117], Copyright (2020) American Chemical Society; (e) is adapted with permission from [28], Copyright (2020) American Chemical Society).

**Figure 11 materials-15-08783-f011:**
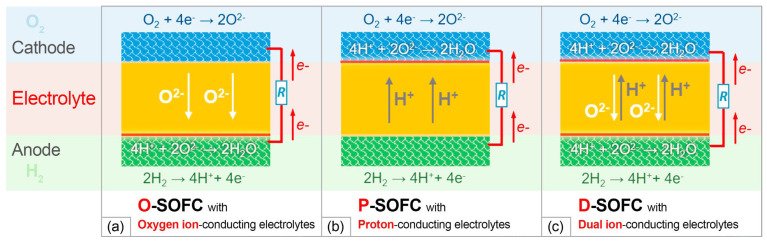
Schematic representation of SOFCs: (**a**) with an oxygen ion-conducting electrolyte; (**b**) with a proton-conducting electrolyte; (**c**) dual-conducting electrolyte (adopted from [135]).

**Figure 12 materials-15-08783-f012:**
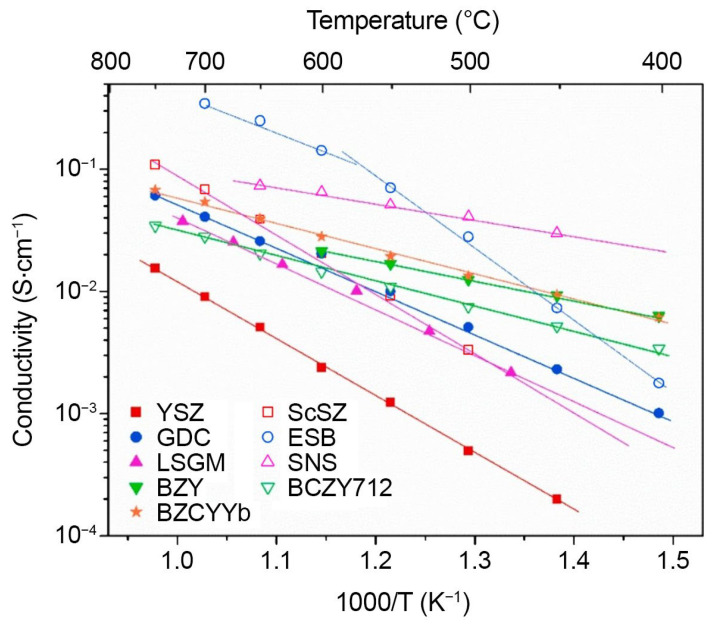
Conductivity of well-established solid-state electrolytes for SOFCs: 8 mol.% Y_2_O_3_–ZrO_2_ (YSZ), 8 mol.% scandia-stabilized ZrO_2_ (ScSZ), Gd_0.1_Ce_0.9_O_1.95_ (GDC), Er_0.4_Bi_0.6_O_3_ (ESB), La_0.8_Sr_0.2_Ga_0.8_Mg_0.2_O_3-δ_ (LSGM), Sr_0.55_Na_0.45_SiO_2.755_ (SNS), BaZr_0.8_Y_0.2_O_3-δ_ (BZY), BaCe_0.7_Zr_0.1_Y_0.2_O_3-δ_ (BCZY712) and BaZr_0.1_Ce_0.7_Y_0.1_Yb_0.1_O_3-δ_ (BZCYYb) (adapted from [135]).

**Figure 13 materials-15-08783-f013:**
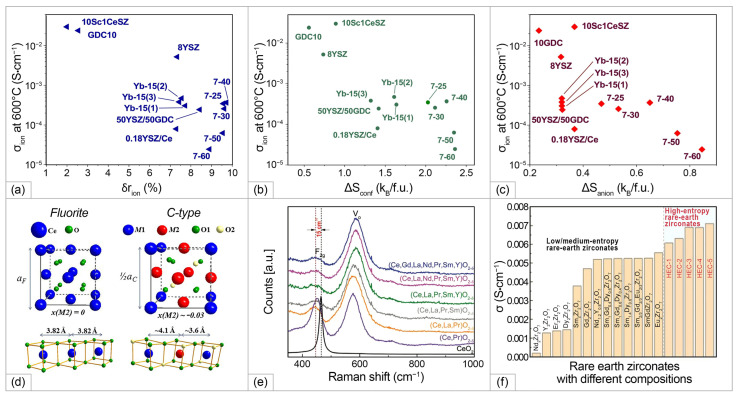
Dependence of the ionic conductivity of some HE and conventional solid oxide electrolytes on (**a**) $\Delta S_{conf}$; (**b**) $\;\delta r_{ion}$; (**c**) $\Delta S_{anion}$; (**d**) schematic representation of fluorite and C-type structures; (**e**) Raman spectra of pure CeO_2_ and different CeO_2_-based rare earth oxides; (**f**) conductivity of low, middle and high-entropy zirconates ((**a**–**c**) are adapted from [165]; (**d**) is adapted and reproduced from [170]; (**e**) is adapted from [119] with permission of The Royal Society of Chemistry; (**f**) is adapted from [171] with permission of Elsevier).

**Figure 14 materials-15-08783-f014:**
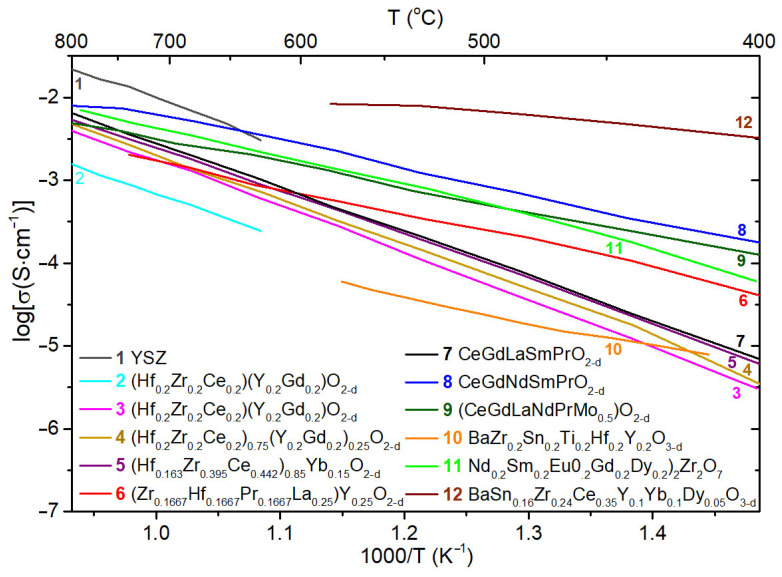
Arrhenius dependences of total conductivity of HEO-based solid-state electrolytes (constructed using the data: 1, 2 [163], 3, 4, 5 [165], 6 [14], 7 [173], 8, 9 [174], 10 [12], 11 [171], 12 [175]).

**Figure 15 materials-15-08783-f015:**
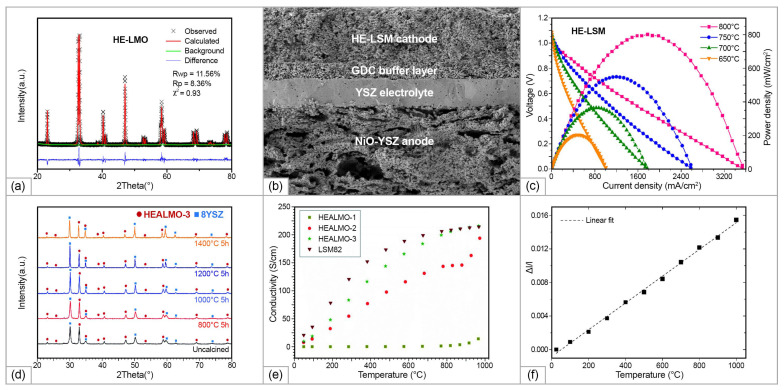
(**a**) XRD Rietveld refinement of LaMn_0.2_Fe_0.2_Co_0.2_Ni_0.2_Cu_0.2_O_3-δ_; (**b**) cross-section image of NiO-YSZ|YSZ|GDC|HE_LSM single cell; (**c**) typical I–V–P plots for NiO-YSZ|YSZ|GDC|HE_LSM single cell; (**d**) XRD patterns of La_0.2_Nd_0.2_Sm_0.2_Ca_0.2_Sr_0.2_MnO_3_ (HEALMO-3)+YSZ mixtures heated at different temperatures for 5 h in air; (**e**) temperature dependencies of electrical conductivity for La_0.2_Nd_0.2_Sm_0.2_Y_0.2_Gd_0.2_MnO_3_ (HEALMO-1), La_0.2_Nd_0.2_Pr_0.2_Sr_0.2_Ba_0.2_MnO_3_ (HEALMO-2), La_0.2_Nd_0.2_Sm_0.2_Ca_0.2_Sr_0.2_MnO_3_ (HEALMO-3), and LSM; (**f**) temperature dependency of relative elongation for La_0.7_Sr_0.3_Co_0.2_Cr_0.2_Fe_0.2_Mn_0.2_Ni_0.2_O_3-δ_ ((**a**) is reproduced from [17] with permission of Elsevier; (**b**,**c**) are reproduced from [179] with permission of Elsevier; (**d**,**e**) are reproduced from [180] with permission The Royal Society of Chemistry; (**f**) is reproduced from [182]).

**Figure 16 materials-15-08783-f016:**
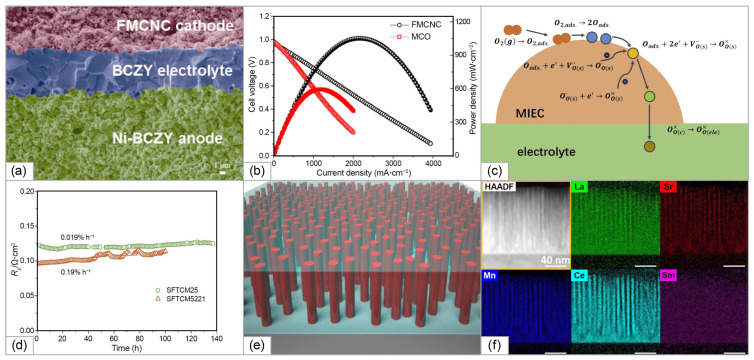
(**a**) Cross-section image of NiO-BCZY|BCZY|GDC|FMCNC single cell; (**b**) fuel cell performances for NiO-BCZY|BCZY|GDC|FMCNC and NiO-BCZY|BCZY|GDC|MCO single cells at 700 °C; (**c**) scheme of oxygen reduction reaction on MIEC cathode; (**d**) polarization resistance of SrFe_0.25_Ti_0.25_Co_0.25_Mn_0.25_O_3-δ_ (SFTCM25) and SrFe_0.5_Ti_0.2_Co_0.2_Mn_0.1_O_3-δ_ (SFTCM5221) cathodes on LSGM electrolyte at 700 °C; (**e**) LSM-SDC VAN structure: schematic 3D-image; (**f**) columnar structure of the film (HAADF, image) and EDS maps of elements distribution in the film ((**a**,**b**) are reproduced from [187]; (**c**,**d**) are reproduced from [188] with permission of Elsevier; (**e**,**f**) are reproduced from [227]).

**Table 1 materials-15-08783-t001:** Performances of SOFCs with electrodes based on high- and medium-entropy oxides.

Anode	Buffer Layer| Electrolyte| Buffer Layer	Cathode	T, °C	MPD, mW/cm^2^	Ref.
NiO-YSZ	|YSZ|CGO	LaMn_0.2_Fe_0.2_Co_0.2_Ni_0.2_Cu_0.2_O_3-δ_	800	551	[17]
NiO-YSZ	|YSZ|CGO	La_0.2_Pr_0.2_Nd_0.2_Sm_0.2_Sr_0.2_MnO_3-δ_	800	801	[179]
NiO-YSZ	|YSZ|GDC	La_0.2_Pr_0.2_Nd_0.2_Sm_0.2_Ba_0.1_Sr_0.1_Co_0.2_Fe_0.6_Ni_0.1_Cu_0.1_O_3-δ_	800	714	[181]
NiO-GDC	GDC|LSGM8282|	La_0.7_Sr_0.3_Co_0.2_Cr_0.2_Fe_0.2_Mn_0.2_Ni_0.2_O_3-δ_	900	550	[182]
NiO-YSZ	|YSZ|GDC	La_0.2_Sr_0.2_Pr_0.2_Y_0.2_Ba_0.2_Co_0.2_Fe_0.8_O_3-δ_	750	1006	[183]
NiO-YSZ	|YSZ|GDC	SmBa(Mn_0.2_Fe_0.2_Co_0.2_Ni_0.2_Cu_0.2_)_2_O_5+δ_	800	684	[185]
NiO-YSZ	|YSZ|GDC	SmBa(Mn_0.2_Fe_0.2_Co_0.2_Ni_0.2_Cu_0.2_)_2_O_5+δ_-GDC	800	839	[185]
NiO-YSZ	|YSZ|CGO	(La_0.2_Pr_0.2_Nd_0.2_Sm_0.2_Gd_0.2_)_2_CuO_4_	700	528	[15]
NiO-BCZY	|BCZY|	Fe_0.6_Mn_0.6_Co_0.6_Ni_0.6_Cr_0.6_O_4_	700	1052	[187]
NiO-CGO	LDC|LSGM8282|	SrFe_0.25_Ti_0.25_Co_0.25_Mn_0.25_O_3-δ_	800	850	[188]

## Data Availability

Not applicable.

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
