# Peer review of "High-Entropy Materials in SOFC Technology: Theoretical Foundations for Their Creation, Features of Synthesis, and Recent Achievements"

_materials, 2022, doi:10.3390/ma15248783_

Round 1

Reviewer 1 Report

Pikalova et al. have established a very nice piece of scientific review article trying to connect the recent achievements in the application of high-entropy alloys and high-entropy oxides in the technology of solid oxide fuel cells. They explained a detailed mechanism of the stabilization of a high-entropy state of the above-mentioned materials including the effect of structural and charge factors on the stability of the resulting homogeneous solid solution. The paper is presented very well and is scientifically sound. This paper could be a good reference for future solid oxide fuel cells. I would recommend this journal to be published in the prestigious MDPI Materials journal. However, I would strongly recommend the authors significantly improve the resolution of the all figures presented in the paper. Good luck!

Author Response

Dear Reviewer! We are grateful to you for the your great attention to our manuscript and its high evaluation.  We have taken into consideration your comment, and we have prepared the high-resolution versions of all figures for download.

 With Best Regards!

 Authors.

Reviewer 2 Report

I think the authors give a sound organization of the literatures in the field of HEA/O materials for SOFCs. This review would contribute significantly to the community. I have only one question:

In Line 117 the authors stated that "The entropy of mixing in a multi-component equiatomic system is largely determined by the configurational entropy". What are the other components of the entropy value? How would the values of these entropies look like when compared to the configurational entropy? It would be fantastic to present an example of entropy calculation of a HEA/O materal system.

Author Response

Dear Reviewer! We are grateful to you for your kind attention to our manuscript and its high evaluation. 

We have taken into consideration your questions, and we made the following changes to the article:

We have changed the definition of configurational entropy at Line 117, and we have redrawn Fig. 3.  We have added the paragraph at Lines 175-191 with the discussion of contributions to the total entropy, and three corresponding references [39-41].

With Best Regards!

Authors.

Reviewer 3 Report

The high-entropy materials in SOFC technology: theoretical foundations for their creation, features of synthesis and recent achievements is shown in this paper. The review highlights of advantages of high-entropy materials, e.g.  high strength and the sluggish diffusion of components, which are promising for the use at the elevated temperatures (operation temperature of SOFCs). In this review the numerous works have been cited, eg on high flexibility in determining and prioritizing directions in the development of high-entropy electrolytes and electrodes for SOFCs operating in the intermediate and low temperature ranges. 

In this work the theoretical aspects and synthesis methods of high-entropy alloys and high-entropy oxides, as well as recent achievements and future prospects for the application of these materials in the technology of SOFC were considered. Attention was also given to the dissimilarity between the stabilization mechanisms of high-entropy alloys  and high-entropy oxides. It was noted in the work that using the methods of atomistic modeling, computational thermodynamics, artificial intelligence methods, and other possible modeling approaches could give a solid base for the creation of new entropy-stabilized SOFC materials with optimized functional properties.

The work contains a citation of many references well matched to the subject of the review. This overview can be very helpful for initial estimation of problems during calculations or laboratory research related to SOFCs. Moreover, this work may contribute to the interest of scientists starting their research on SOFCs.

References well matched to the topic.

Review well described.

Some suggestion follows:

- it should be good to rewrite and condensed the conclusions,

- it should be good to increase the resolution of the figures - to help the readers understand the topic.

Author Response

Dear Reviewer! We are grateful to you for your serious attention to our manuscript, for your kind review, and for your high evaluation of our work. We have taken into consideration your kind comments, and we have rewritten and emphasized the conclusions. Besides, we have prepared the high-resolution versions of all figures for download.

With Best Regards!

Authors.

Reviewer 4 Report

This work is interesting. Although, there are several comments required to be undertaken by the authors before it could be accepted.

1.     The abstract is too general, I suggest that you should just provide brief intro of the issues, while mainly focusing on the summary of the works and results reviewed/obtained or analyzed.

2.   Instead of reporting applications, I suggest that authors should critically analyzed the previous literatures for what has been done, research gaps and limitations to justify the novelty of this review work.

3.  The sub-section 3.3.1 and 3.3.2 are too long, try and synthesize the "message" of your findings to place the work in context.

4.    There is no in-depth analysis for published works related to application of hydrogen/fuel cells in vehicles. Kindly add 10.1016/j.jclepro.2022.134238 reference to support your work.

5.    Add future prospects of your investigations at the end of discussion.

Author Response

Dear Reviewer! We are grateful to you for your attention to our manuscript, and we thank you for your comments.

We have taken into consideration your comments, and we made the following changes to the manuscript:

  1. We have rewritten Abstract with the aim to do it more accented and productive.
  2. We would like to emphasise that our Review is focused, aim of it is to present the recent applications of high-entropy materials (HEMs) as constructional parts of solid oxide fuel cells (SOFCs), that is why, using of HEMs in SOFCs stands in our manuscript at the first place. And, of course, we have accented in the text the attention of readers on disadvantages and limitations of HEMs during their using in SOFCs.
  3. We have shortened the Section 3.3.2 (after Line 805), and Section 3.3.3 (Lines 1005, 1019, 1056, 1104, 1120, 1134, 1152, 1176, 1219, 1237), deleting the excess information. 
  4. We have added the proposed reference, [5].
  5. We have added future prospects of our investigations into Conclusions Part owing to your comments.

With Best Regards!

Authors.